# Unlocking the Power of Representations in Long-term Novelty-based Exploration

**Alaa Saade**[*]**, Steven Kapturowski**[*]**, Daniele Calandriello**[*]**, Charles Blundell,**
**Pablo Sprechmann, Leopoldo Sarra**[†]**, Oliver Groth, Michal Valko, Bilal Piot.**
Google Deepmind
```
{alaas,skapturowski,dcalandriello,
 cblundell,psprechmann,leopoldo.sarra,ogroth,valkom,piot}@google.com
```

## Abstract

We introduce *Robust Exploration via Clustering-based Online Density Estimation* (`RECODE`), a non-parametric method for novelty-based exploration that estimates visitation counts for clusters of states based on their similarity in a chosen embedding space. By adapting classical clustering to the nonstationary setting of Deep RL, `RECODE` can efficiently track state visitation counts over thousands of episodes. We further propose a novel generalization of the inverse dynamics loss, which leverages masked transformer architectures for multi-step prediction; which in conjunction with `RECODE` achieves a new state-of-the-art in a suite of challenging 3D-exploration tasks in `DM-HARD-8`. `RECODE` also attains state-of-the-art performance in hard exploration Atari games, and is the first agent to reach the end screen in *Pitfall!*

## 1 Introduction

Exploration mechanisms are a key component of reinforcement learning (RL, Sutton & Barto, 2018) agents, especially in sparse-reward tasks where long sequences of actions need to be executed before collecting a reward. The exploration problem has been studied theoretically (Kearns & Singh, 2002; Azar et al., 2017; Brafman & Tennenholtz, 2003; Auer et al., 2002; Agrawal & Goyal, 2012; Audibert et al., 2010; Jin et al., 2020) in the context of bandits (Lattimore & Szepesvári, 2020) and Markov Decision Processes (MDPs, Puterman, 1990; Jaksch et al., 2010). One simple yet theoretically-sound approach for efficient exploration in MDPs is to use a decreasing function of the visitation counts as an exploration bonus (Strehl & Littman, 2008; Azar et al., 2017). However, this approach becomes intractable for large or continuous state spaces, where the agent is unlikely to visit the exact same state multiple times, and some form of meaningful generalization over states is necessary. Several approximations and proxies for visitation counts and densities have been proposed to make this form of exploration applicable to complex environments. Two partially successful approaches in deep RL are: the parametric approach, which uses neural networks to estimate visitation densities directly, and the non-parametric approach, which leverages a memory of visited states to guide exploration.

Parametric methods either explicitly estimate the visitation counts using density models (Bellemare et al., 2016; Ostrovski et al., 2017) or use proxies for visitation such as the prediction error of a dynamics model (Pathak et al., 2017; Guo et al., 2022), or from predicting features of the current observation, e.g., features given by a fixed randomly initialized neural network as in `RND` (Burda et al., 2019). While this family of methods provides strong baselines for exploration in many settings (Burda et al., 2018), they are prone to common problems of deep learning in continual learning scenarios, especially as slow adaptation and catastrophic forgetting. Parametric models trained via gradient descent are generally unsuitable for rapid adaptation (e.g., within a single episode) because it requires updates to the state representation before the exploration bonus can catch up. Additionally, catastrophic forgetting makes parametric methods susceptible to the so-called 'detachment' problem in which the algorithm loses track of promising areas to explore (Ostrovski et al., 2017). Non-parametric methods rely on a memory to store encountered states (Savinov et al., 2018; Badia et al., 2020b). This facilitates responsiveness to the most recent experience as well as preserving memories without interference. However, due to computational constraints, it is necessary to limit the memory size which, in turn, requires a selection or aggregation mechanism for states.

---

[*]Equal contributions, [†] Department of Physics, Friedrich-Alexander Universität Erlangen-Nürnberg, work done while interning at DeepMind.

To obtain the best of both worlds, Never Give Up (`NGU`, Badia et al., 2020b) combines a short-term novelty signal based on an episodic memory and a long-term novelty signal based on `RND` into a single intrinsic reward. However, the need to estimate two different novelty signals simultaneously adds complexity and requires careful tuning. Moreover, as pointed out by Pathak et al. (2017), the final efficacy of any exploration algorithm strongly depends on the chosen state representation. If the state encoding is susceptible to noise or uncontrollable features in the observations, it can lead to irrelevant novelty signals and prevent meaningful generalization over states. As `NGU` relies on `RND` for representation, it also inherits its encoding deficiencies in the presence of noisy observations which limits the applicability of the method in stochastic or complex environments.

In this paper, we tackle these issues by decomposing the exploration problem into two disentangled sub-problems. First, (i) **Representation Learning** with an embedding function that encodes a meaningful notion of state similarity while being robust to uncontrollable factors in the observations. Second, (ii) **Count Estimation** that is able to provide a long term visitation-based exploration bonus while retaining responsiveness to the most recent experience. Addressing (i), we extend the inverse dynamic model proposed by Pathak et al. (2017) by leveraging the power of masked sequence transformers (Devlin et al., 2018) to build an encoder which can produce rich representations over longer trajectories while suppressing the encoding of uncontrollable features. We refer to our representation as `CASM`, for *Coupled Action-State Masking*. In order to deliver on (ii) we introduce a novel,

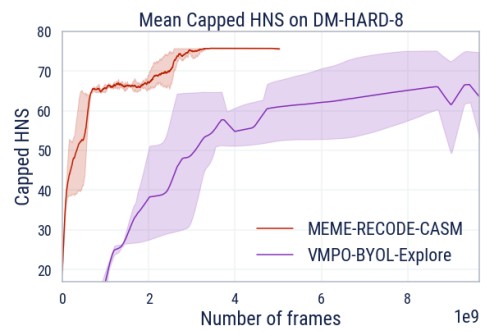

Figure 1: A key result of `RECODE` is that it allows us to leverage more powerful state representations for long-term novelty estimation. This enables new state-of-the-art performances in the challenging 3D task suite `DM-HARD-8`, where for the first time we achieve super-human performance (i.e., exceed 100 in human normalized score) in 6 out of 10 tasks.

non-parametric method called Robust Exploration via Clustering-based Online Density Estimation (`RECODE`). In particular, `RECODE` estimates soft visitation counts in the embedding space by adapting density estimation and clustering techniques to an online RL setting. Our approach tracks histories of interactions spanning thousands of episodes, significantly increasing memory capacity over prior art in non-parametric exploration methods which typically only store the most recent history like the current episode. In the presence of noise, we show that it strictly improves over state-of-the-art exploration bonuses such as `NGU` or `RND`. `RECODE` matches or exceeds state-of-the-art exploration results on Atari and is the first agent to reach the end-screen in *Pitfall!*, a notoriously difficult task due to strict in-game time limits that require discovering an efficient route that explores and backtracks across 255 rooms. Beyond 2D, our method also performs well in much harder 3D domains and in conjunction with `CASM`, sets new state-of-the-art results in the challenging `DM-HARD-8` suite (Fig. 1) in terms of human normalized score (HNS, Mnih et al. (2015)).

## 2 BACKGROUND

We consider a discrete-time interaction (McCallum, 1995; Hutter, 2004; Hutter et al., 2009; Daswani et al., 2013) between an agent and its environment. At each time step $t \in \mathbb{N}$ the agent receives an observation $o_t \in \mathcal{O}$, that partially captures the underlying state $s \in \mathcal{S}$ of the environment and generates an action $a_t \in \mathcal{A}$. We consider policies $\pi : \mathcal{O} \to \Delta_{\mathcal{A}}$, that map an observation to a probability distribution over actions. Finally, an extrinsic reward function $r_e : \mathcal{S} \times \mathcal{A} \to \mathbb{R}$ maps an observation to a scalar feedback. This function can be combined with an intrinsic reward function $r_i$ to encourage the exploratory behavior which might not be induced from $r_e$ alone.

The observations provided to the agent at each time step $t$ are used to build a representation of the state via an embedding function $f_\theta : \mathcal{O} \to \mathcal{E}$, associating $o_t$ with a vector $e_t = f_\theta(o_t)$. Typically, the embedding space $\mathcal{E}$ is the vector space $\mathbb{R}^D$ where $D \in \mathbb{N}^*$ is the embedding size. Common approaches to learn $f_\theta$ include using an auto-encoding loss on the observation $o_t$ (Burda et al., 2018), an inverse dynamics loss (Pathak et al., 2017), a multi-step prediction loss at the latent level (Guo et al., 2020; 2022), or other similar representation learning methods. In particular, Pathak et al. (2017) and Badia et al. (2020b) highlight the utility of the inverse-dynamics loss to filter out noisy or uncontrollable features, e.g., an on-screen death timer as in *Pitfall!*.

A popular and principled approach to exploration in discrete settings is to provide an intrinsic reward inversely proportional to the visitation count (Strehl & Littman, 2008; Azar et al., 2017). However, in large or continuous spaces the same state may be rarely encountered twice. Badia et al. (2020b) remedy this issue by introducing a slot-based memory $M$, which stores all past embeddings in the current episode, and replaces discrete counts with a sum of similarities between a queried embedding $e_t = f_\theta(o_t)$ and its k-nearest-neighbors $\text{Neigh}_k(e_t)$ under the kernel $\mathcal{K}$:

$$r_t \propto \frac{1}{\sqrt{N(f_\theta(o_t))}} \approx \frac{1}{\sqrt{\sum_{m \in \text{Neigh}_k(e_t)} \mathcal{K}(e_t, m)}}. \tag{1}$$

Since storing the full history of embeddings throughout training would require a prohibitive amount of space, this slot-based memory is typically relegated to short-term horizons only, and in `NGU` it is reset at the end of every episode. As a consequence, slot-based memory must be combined with a separate mechanism capable of estimating long-term novelty; resulting in additional method complexity and trade-offs. In the following, we present a simple and efficient slot-based memory which can effectively track novelty over thousands of episodes.

## 3 RECODE

We will now introduce our method, Robust Exploration via Clustering-based Online Density Estimation (`RECODE`), to compute intrinsic rewards for exploration. `RECODE` takes inspiration from the reward of `NGU` (Badia et al., 2020b), but while `NGU` stores individual embedded observations in $M$ and uses periodic resets to limit space complexity, `RECODE` controls its space complexity by aggregating similar observations in memory. This requires storing a separate counter associated with each element in the memory and new observations need not be directly added to the memory, but will typically be assigned to the nearest existing element whose counter is then incremented. Since the counters are never reset and the merged observations have a better coverage of the embedding space, `RECODE`'s memory is much longer-term than a simple slot-based approach, yielding state-of-the-art performance in many hard-exploration tasks. It also simplifies the estimation of novelty to only one mechanism vs. two as in `NGU`. Moreover, the `RECODE` architecture is highly flexible, allowing it to be easily combined with a variety of RL agents and most importantly different representation learning methods. As we show in the experiments, methods that can better leverage priors from learned representations, such as `RECODE`, outperform those that need to estimate novelty directly on raw observations, like `RND` (and in turn `NGU`). We now present more in detail `RECODE`, summarized in. Alg. 1.

**Approximating visitation counts.** Our estimator is based on a finite slot-based container $M = \{m_j\}_{j=1}^{|M|}$, where $|M|$ is the memory size. We refer to $m_j \in \mathcal{E}$ as atoms since they need not correspond to a single embedding as in Badia et al. (2020b;a) We also store a separate count vector $c$ such that $c_i$ is an estimate of the visitation count of $m_i$. In particular, $c_i$ does not only reflect the number of visits to $m_i$ but also captures any previous visit sufficiently close to it.

Given a new embedding $e$, we estimate its *soft-visitation count* (Alg. 1:L3-4) as the weighted sum of all atoms close to $e$ in the memory, according to a similarity kernel:

$$N_\mathcal{K}(M, e) = \sum_l (1 + c_l) \mathcal{K}(m_l, e; d_{\text{ema}}). \tag{2}$$

In particular, we choose our kernel function as:

$$\mathcal{K}(m_l, e) = \frac{1}{1 + \frac{\|e - m_l\|_2^2}{\epsilon d_{\text{ema}}^2}} \, \mathbb{1}_{\left\{\|e - m_l\|_2^2 < d_{\text{ema}}^2\right\}}, \tag{3}$$

where $\epsilon \in \mathbb{R}_+$ is a fixed parameter. Eq. (3) is similar to Badia et al. (2020b), but we replace their sum over $e$'s top-$k$ neighbors with a sum over all atoms within a $d_{\text{ema}}$ distance from $e$. This choice prevents a counter-intuitive behaviour that can occur when using the $k$-NN approach with counts. In particular, it is desirable that the soft-visitation count of a given embedding should increase after adding it to the memory. However, adding atoms to the memory can change the $k$-NN list. If an atom displaced from this list has a large count, this might actually *reduce* nearby soft-visitation count estimates instead of increasing them. Conversely, our approach is not affected by this issue.

Finally, we return $r$ as in Eq. (1), but add a small constant $n_0$ to the denominator for numerical stability and normalize $r$ by a running estimate of its standard-deviation as in Burda et al., 2019.

---

**Algorithm 1** `RECODE`

---

1: **Input:** Embedding $e$, Memory $M = \{m_l\}_{l=1}^{|M|}$, atom visitation counts $\{c_l\}_{i=l}^{|M|}$, number of neighbors $k$, relative tolerance to decide if a candidate new atom is far $\kappa$, squared distance estimate $d_{\text{ema}}^2$, $d_{\text{ema}}^2$'s decay rate $\tau$, discount $\gamma$, insertion probability $\eta$, kernel function $\mathcal{K}$, intrinsic reward constant $n_0$
2: **Output:** Updated memory $M = \{m_l\}_{l=1}^{|M|}$, updated atom visitation counts $\{c_l\}_{i=l}^{|M|}$, updated squared distance $d_{\text{ema}}^2$, intrinsic reward $r$
3: Compute $N_\mathcal{K}(M, e) = \sum_{l=1}^{|M|} (1 + c_l) \, \mathcal{K}(m_l, e)$;
4: Compute intrinsic reward $r = \left( \sqrt{N_\mathcal{K}(M, e)} + n_0 \right)^{-1}$
5: Find nearest $k$ atoms to the embedding $e$: $\text{Neigh}_k(e) = \{m_j\}_{j=1}^k$
6: Update $d_{\text{ema}}$ estimate: $d_{\text{ema}}^2 \leftarrow (1 - \tau) \, d_{\text{ema}}^2 + \frac{\tau}{k} \sum_{m \in \text{Neigh}_k(e)} \|m - e\|_2^2$
7: Discount all atom counts $c_l \leftarrow \gamma \, c_l \quad \forall l \in \{1, \cdots, |M|\}$
8: Find nearest atom $m_\star = \arg\min_{m \in M, m \neq m_j} \|m - e\|_2$
9: Sample uniformly a real number in $[0, 1]$: $u \sim U[0, 1]$
10: **if** $\|m_\star - e\|_2^2 > \kappa \, d_{\text{ema}}^2$ and $u < \eta$ **then**
11:      Sample atom to remove $m_j$ with probability $P(j) \propto 1/c_j^2$
12:      Find atom $m_\dagger$ nearest to $m_j$: $m_\dagger = \arg\min_{m \in M, m \neq m_j} \|m - m_j\|_2$
13:      Redistribute the count of removed atom: $c_\dagger \leftarrow c_j + c_\dagger$
14:      Insert $e$ at index $j$ with count 1: $m_j \leftarrow e \, , c_j \leftarrow 1$
15: **else**
16:      Update nearest atom position $m_\star \leftarrow \frac{c_\star}{c_\star + 1} m_\star + \frac{1}{c_\star + 1} e$
17:      Update nearest atom count $c_\star \leftarrow c_\star + 1$
18: **end if**

---

**Building the memory.** To build our memory we rely on the same aggregation principle we used to estimate soft-visitation counts, drawing a parallel between our atoms $m_i$ and the centroids of a clustering of observations. We take inspiration from classical clustering and density estimation approaches such as $k$-means or DP-means Kulis & Jordan (2011); and adapt them to deal with the challenges posed by our large scale RL setting: memory size is limited and cannot store all past data, observations arrive sequentially, their distribution is non-stationary, and even the representation used to embed them changes over time. We now describe how `RECODE` tackles these problems.

At every step we must update the memory $M$ to reflect the impact of seeing $e$ on the soft-visitation counts, while keeping the size $|M|$ fixed. Intuitively, two possible ways come to mind: either replace an existing atom with the new embedding, or update the position and count of an existing atom to be closer to $e$. Let $m_\star$ be the closest atom to $e$ in $M$. We adopt the following rules (Alg. 1:L8-18) to integrate new embeddings into the memory, which are closely related to the DP-means clustering algorithm Kulis & Jordan (2011):

- If $e$ satisfies $\|m_\star - e\|^2 < \kappa d_{\text{ema}}^2$, where $d_{\text{ema}}$ is an adaptive threshold and $\kappa > 0$ a fixed parameter, it is "assigned" to the cluster encoded by $m_\star$ and we update $m_\star$'s value according to the convex combination of the counts of the existing embedding and the new one:

$$m_\star \longleftarrow \frac{c_\star}{c_\star + 1} m_\star + \frac{1}{c_\star + 1} e \tag{4}$$

  Its weight $c_\star$ is also incremented by 1;

- If there is no close-by atom, we randomly decide whether to create a new one by flipping a coin with probability $\eta$. If the coin-flip succeeds, we introduce the new embedding as a new atom, and we also remove an existing atom using a procedure described in the next paragraph. If the coin-flip fails, we instead update $m_\star$ as in equation 4.

The random coin-flip is introduced to increase the stability of the clustering algorithm to noise. In particular, an embedding far away from the memory will be inserted only after it is seen on average $1/\eta$ times, making one-off outliers less of a problem. At the same time, once a far away embedding is observed multiple times and becomes relevant for the soft-visitation counts, there is a high chance that it will be added to improve the coverage of the memory. But to keep memory size finite, an existing atom must be removed. We investigate three different strategies to select an atom $m_i$ for removal

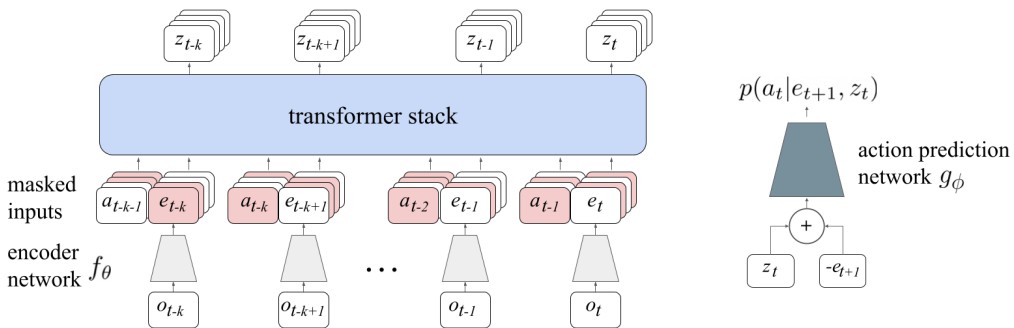

Figure 2: Coupled Action-State Masking (CASM) architecture used for learning representations in partially observable environments. The transformer takes masked sequences of length $k$ consisting of actions $a_i$ and embedded observations $e_i = f_\theta(o_i)$ as inputs and tries to reconstruct the missing embeddings in the output. The reconstructed embeddings at time $t-1$ and $t$ are then used to build a 1-step action-prediction classifier. The embedding function used as a representation for RECODE is $f_\theta$. Masked inputs are shaded in pink, $N = 4$ masked sequences are sampled during training (indicated by the stacks of $a$, $e$ and $z$ in the diagram).

based on its cluster count $c_i$: (a) removing with probability $\propto \frac{1}{c_i^2}$; (b) removing with probability $\propto \frac{1}{c_i}$; (c) removing the atom with the smallest $c_i$. An ablation study over removal strategies in App. D.2 (Figures 8 and 9), empirically shows that strategy (a) works best for the settings we consider, but also that results are generally quite robust to the specific choice.

Whenever an atom $i$ is removed, its count $c_i$ is redistributed to the count of its nearest neighbor in order to preserves the total count of the memory. The update rule of RECODE can be also interpreted from the theoretical point of view as an approximate inference scheme in a latent DP-means probabilistic clustering model. We provide a more detailed connection in App. D.

**Dealing with non-stationary distributions.** The distance scale between embedded observations can vary considerably between environments and throughout the course of training, as a result of non-stationarity in both the policy and embedding function $f_\theta$. To deal with this issue, we include an adaptive bandwidth mechanism as in NGU Badia et al. (2020b). In particular, we update the kernel parameter $d_{\text{ema}}^2$ whenever a new embedding $e$ is received, based on the mean squared distance of the new embedding to the $k$-nearest existing atoms (Alg. 1:L5-6). To allow for faster adaptation of $d_{\text{ema}}$, we replace the running average used in NGU with an exponential moving average with parameter $\tau$.

We note, however, that this mechanism is insufficient to cope with non-stationarity in $f_\theta$ over long timescales. The original NGU memory is not strongly impacted by this issue since it is reset after every episode, leaving little time for the representation to change significantly. However, in RECODE, these changing representations can end up corrupting the long-term memory if old clusters are not updated frequently. In particular, an atom might achieve a high count under a representation, but become unreachable (and thus useless) under a different representation while still being unlikely to be removed. To counteract this we add a decay constant $\gamma$ which discounts the counts of all atoms in memory at each step as $c_i \longleftarrow \gamma c_i$, with $\gamma < 1$ (Alg. 1:L7).

This effectively decreases the counts of stale atoms over time and increases the likelihood of their removal during future insertions: clusters that do not get new observations 'assigned' to them for a long time are eventually replaced. At the same time, relevant clusters are kept alive much longer than previous methods. Fig. 3 reports the histogram of cluster ages for clusters contained in the memory of an agent that has learned how to reach *Pitfall!*'s end screen. The red line in Fig. 3 denotes the maximum possible

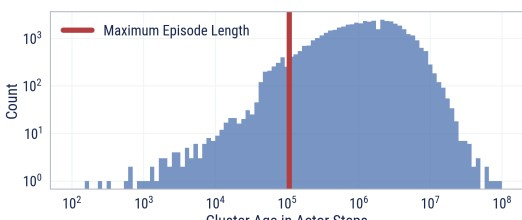

Figure 3: Content of an agent memory when it learns to reach *Pitfall!*'s end screen.

number of steps in an single episode, which is enforced by *Pitfall!*'s in-game death timer, and would represent the maximum memory horizon for methods that reset their memory every episode. As we

can see, most of the clusters are much older than one episode, with earliest memories reaching back thousands of episodes. We consider the effect of discounting in more detail in App. D.2 (Figures 10 to 12 and 14). Importantly, we note that unlike `NGU` where each actor maintains its own copy of the memory, `RECODE` shares the memory across all actors in a distributed agent, which greatly increases the frequency of updates to each atom resulting in less representation drift between memory updates.

**Tuning `RECODE`.**    While we introduced Alg. 1 in its most general form, we observe experimentally that performance is robust w.r.t. most of the hyper-parameters introduced (see App. L). In particular, we note that the choice of discount $\gamma$ and memory size have the largest impact on performance. All other hyper-parameters were chosen via coarse independent sweeps on two to three values and held constant across all experiments (see Sec. 5 and App. F for more details).

## 4    REPRESENTATION LEARNING METHODS

As discussed in Section 2, the choice of the embedding function $f_\theta : \mathcal{O} \to \mathcal{E}$ can have a significant impact on the quality of exploration; with many different representation learning techniques being studied in this context (Burda et al., 2018; Guo et al., 2020; 2022; 2021; Erraqabi et al., 2021). In the following, we focus on action prediction embeddings, introducing first the standard 1-step prediction formulation (Pathak et al., 2017; Badia et al., 2020b;a). The embedding function $f_\theta$ is parameterized as a feed-forward neural network taking $o_t$, the observation at time $t$, as input. We define a classifier $g_\phi$ that, given the embeddings of two consecutive observations $f_\theta(o_t), f_\theta(o_{t+1})$, outputs an estimate $p_{\theta,\phi}(a_t|o_t, o_{t+1}) = g_\phi\left(f_\theta(o_t), f_\theta(o_{t+1})\right)$ of the probability of taking an action given two consecutive observations $(o_t, o_{t+1})$. Both $f_\theta$ and $g_\phi$ are then jointly trained by minimizing an expectation of the negative log likelihood:

$$\min_{\theta,\phi} \mathcal{L}(\theta, \phi)(a_t) = -\ln(p_{\theta,\phi}(a_t|o_t, o_{t+1})), \tag{5}$$

where $a_t$ is the true action taken between $o_t$ and $o_{t+1}$. These embeddings proved to be helpful in environments with many uncontrollable features in the observation (Badia et al., 2020b), such as in Atari's *Pitfall!*, where the observations contain many spurious sources of novelty even when the agent is standing still.

While `RECODE` can be used with an arbitrary embedding function, e.g. one tailored for the domain of interest, the choice of a meaningful representation is also a key factor for the final performance. A major downside of the standard, 1-step action-prediction method is the simplicity of the prediction task, which can often be solved by learning highly localized and low-level features (e.g. how a single object shifts under a transition), which need not be informative of the global environment structure. In contrast, an ideal embedding should capture higher-level information about the environment, such as the agent's position or relative location of previously observed landmarks; which might not be simultaneously present in the individual observations $o_t$ and $o_{t+1}$. In order to achieve this, a wider context of time-steps may be needed.

However, the prediction task would become even easier if we simply provided the full trajectory to the predictor. In order to address this limitation, we propose to use a *stochastic* context, $h_t$, where at each timestep $k \leq t$, either $f_\theta(o_k)$ or $a_{k-1}$ is provided.[1] The main intuition being that the model can still predict $a_t$ by learning to infer the missing information from $f_\theta(o_t)$ given $(h_{t-1}, a_{t-1})$. In this way, the action predictor would not solely rely on the information provided by $f_\theta(o_t)$, but it would also construct redundant representations within $h_t$.

From an implementation standpoint, we first build a sequence of observation embeddings and actions, $(f_\theta(o_0), a_0, f_\theta(o_1), \ldots, a_{t-1}, f_\theta(o_t))$. Then, inspired by masked language models (Devlin et al., 2018), at each timestep $t$, we randomly substitute either $f_\theta(o_t)$ or $a_t$ with a special token indicating missing information. These masked sequences are then fed to a causally-masked transformer, whose output is then projected down to the size of the embedding ($\dim z_t = \dim f_\theta(o_t)$), and the difference between the two is input into a final MLP classifier $g_\phi$. As with 1-step action prediction, we train the representation using maximum likelihood. We refer to this approach as Coupled Action-State Masking (`CASM`) in the following. During training, we randomly sample multiple masked sequences per trajectory ($N = 4$) to help reduce gradient variance. Note that the final embedding that we

---

[1]We avoid masking both $f_\theta(o_k)$ and $a_{k-1}$ simultaneously as this would increase the likelihood that the prediction task is indeterminable.

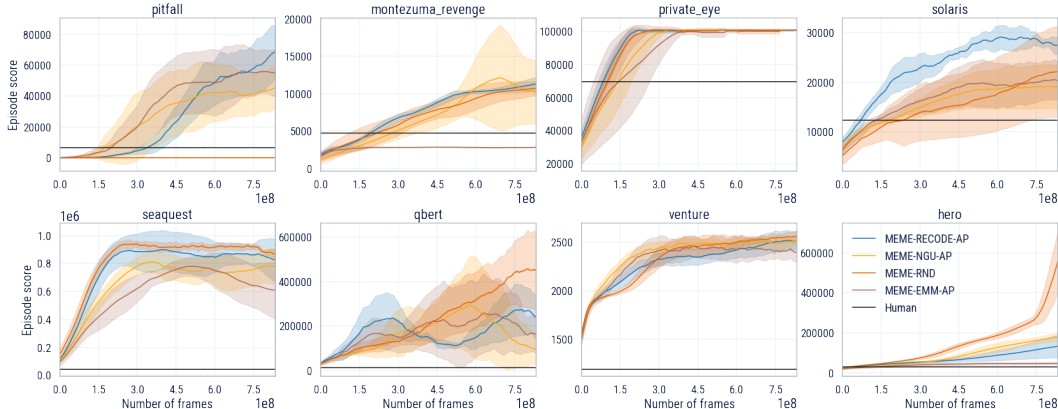

Figure 4: Comparison of `RECODE` against other exploration bonuses on Atari's hard exploration games. All agents are based on `MEME` and use the same representation learning mechanism (`AP`). Note that the high variance in *Q\*bert* is due to a bug in the game that, when exploited, allows to obtain significantly higher scores (Chrabaszcz et al., 2018).

provide to `RECODE` is $e_t = f_\theta(o_t)$, i.e. the transformer *inputs*, to avoid leaking information about the agent's trajectory. Figure 2 shows a diagram of the architecture.

## 5 EXPERIMENTS

In this section, we experimentally validate the efficacy of our approach on two established benchmarks for exploration in 2D and 3D respectively: a subset of the Atari Learning Environment (ALE, Bellemare et al., 2013) containing eight games such as `Pitfall` and `Montezuma's Revenge` which are considered hard exploration problems (Bellemare et al., 2016); and `DM-HARD-8` (Gulcehre et al., 2019), a suite of partially observable 3D games. All games pose significant exploration challenges such as very long horizons ($\mathcal{O}(10K)$ steps), the necessity to backtrack, sparse rewards, object interaction and procedural environment generation. Our method achieves state-of-the-art results across both benchmarks and even solves two previously unsolved games: in Atari's *Pitfall!* our method is the first to reach the end screen and on `DM-HARD-8`'s *Push Block* we are the first to achieve super-human performance. We also perform a set of ablations to shed more light on the influence of the representation learning mechanism and the robustness w.r.t. noisy observations.

All candidate architectures evaluated in the following experiments (and in App. L), are composed of three main modules: (1) a base agent, responsible for core RL tasks such as collecting observations and updating the policy, (2) an algorithm responsible for generating the exploration bonus, and (3) an embedding mechanism responsible for learning meaningful representations of observations. Our nomenclature reflects the choice of modules as `AGENT-EXPLORATION-EMBEDDING`. For example, the `MEME` agent described in Kapturowski et al. (2022) is denoted as `MEME-NGU-AP`. We use the `MEME` agent across all experiments, but vary the exploration and representation mechanisms. For exploration we consider `EMM` (pure episodic memory), `NGU` and `RECODE` whereas for representation we experiment with `AP` and `CASM`. We provide a full list of hyper-parameters for all agents and baselines in App. F.

### 5.1 ATARI

The hard-exploration subset of Atari as identified by Bellemare et al. (2016) poses a considerable challenge in terms of optimization horizon with episodes lasting up to $27,000$ steps using the standard action-repeat of four. Additionally, rewards vary considerably in both scale and density. Across all our experiments in the Atari domain, we set the memory size of our agent to $5 \cdot 10^4$ atoms. We evaluate all agents following the regime established in prior work (Mnih et al., 2015; Van Hasselt et al., 2016) using 30 random no-ops, no 'sticky actions' (Machado et al., 2018) and average performance over 6 seeds. We compare the game scores obtained using our exploration bonus, `RECODE`, against other methods while keeping agent architecture and representation mechanism fixed. The results presented in Fig. 4 show that our method achieves state-of-the-art, super-human performance across all eight games while using a conceptually simpler exploration bonus compared to `MEME-NGU-AP`. The `MEME-EMM-AP` and `MEME-RND` ablations in Fig. 4 reveal the respective shortcomings of short-term

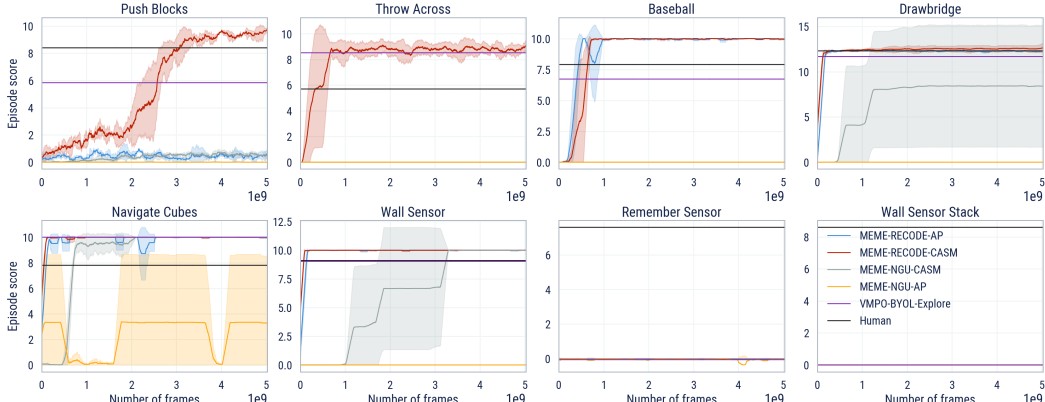

Figure 5: Performance of `RECODE` compared to `NGU` and `BYOL-Explore` on the single-task version of `DM-HARD-8`. The BYOL-Explore results correspond to the final performance reported in Guo et al. (2022) after `1e10` environment frames. All results have been averaged over 3 seeds.

and long-term novelty when used in standalone fashion. `EMM` on its own cannot solve *Montezuma's Revenge* because it requires long-term memory. Conversely, `RND` on its own cannot solve *Pitfall!* because of the presence of many uncontrollable features in the observations and its inability to leverage the `AP` embeddings. In contrast, `RECODE` is able to leverage the `AP` representation for short-term and long-term novelty due to the clustering-based memory integrating over a long horizon which enables solving both games with a single intrinsic reward.

## 5.2 `DM-HARD-8`

`DM-HARD-8` (Gulcehre et al., 2019) consist of eight exploration tasks, designed to challenge an RL agent in procedurally-generated 3D worlds with partial observability, continuous control, sparse rewards, and highly variable initial conditions. Each task requires the agent to interact with specific objects in its environment in order to reach a large apple that provides reward (cf. Fig. 16 in the Appendix for an example). The procedural generation randomizes object shapes, colors, and positions at every episode. Across all our experiments in the `DM-HARD-8` domain, we set the memory size of our agent to $2 \cdot 10^5$ atoms. We also use the more powerful `CASM` representation over `AP` as the default in these experiments but present an ablation on the influence of the representation in Sec. 5.3. All performances reported for evaluation are averaged across three seeds.

We compare `RECODE` with `NGU` and the recently proposed `BYOL-Explore` (Guo et al., 2022) in this domain. The results presented in Fig. 5 show that our method is able to solve six out of eight tasks with super-human performance which sets a new state-of-the-art on this benchmark and marks the first time that the human baseline has been beaten on *Push Blocks*. To control for the contribution of the representation, we also run a version of `NGU` which uses the more powerful `CASM` representation instead of its default `AP` one. Switching `AP` with `CASM` improves `NGU`'s performance significantly and stresses the importance of incorporating information over longer trajectories in the representation mechanism for this domain to combat the challenge of partial observability. However, only `RECODE` is able to take full advantage of the representational power afforded by `CASM` as it is able to leverage it for both short-term and long-term novelty bonuses.

## 5.3 ABLATIONS

Concluding our experiments, we perform two ablation studies to gauge the sensitivity of our approach to the presence of noisy observations and the choice of the underlying representation mechanism.

**Robustness to observation noise.** Noise in the observation space is one of the most significant adversarial conditions exploration methods must to overcome to deliver utility for any practical scenario which always features imperfect sensors. The 'noisy TV problem' (Schmidhuber, 2010; Pathak et al., 2017) is a common metaphor which describes a failure mode of exploration methods getting stuck on the prediction of noise as a meaningless signal of novelty. In order to assess our method's robustness w.r.t. observation noise, we construct a noisy version of *Montezuma's Revenge*

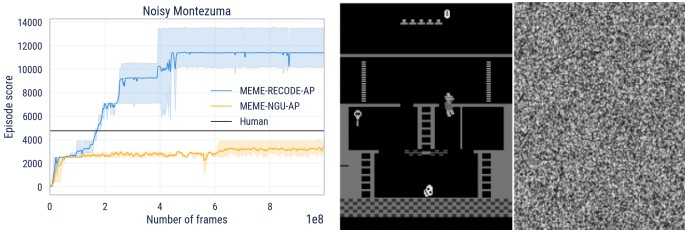 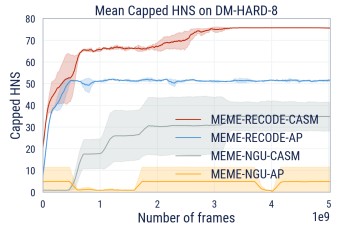

Figure 6: Robustness to observation noise. Top: Performance of `RECODE` compared to `NGU` on Noisy Montezuma. Bottom: A frame of Noisy Montezuma where the noise is concatenated to the original frame.

Figure 7: Comparing `AP` to `CASM` on `DM-HARD-8` for both `RECODE` and `NGU`.

by concatenating a frame containing white noise in the range $[0, 255]$ to the game's original $210 \times 160$ greyscale observations along the image height dimension. We compare `RECODE` to `NGU` in this setting using the same `AP` backbone to suppress uncontrollable noise on the representation level and assess the sensitivity of the exploration bonus to it. The results of this experiment are presented in Fig. 6. We find that the performance of `MEME-NGU-AP` deteriorates significantly in the presence of noise. This can be attributed to the fact that `NGU` relies on `RND` to compute the long-term exploration bonus, which degenerates to random exploration in the presence of uncontrollable noise (Kapturowski et al., 2018). This effectively restricts the baseline to short-term exploration within one episode. In contrast, `RECODE`'s mean performance is not degraded significantly and achieves a similar score as in Fig. 4, albeit with a higher variance.

**Leveraging different representation mechanisms.** The experiments on `DM-HARD-8` demonstrate the importance of employing more powerful representation learning techniques in more complex, partially observable environments. However, while a richer representation often provides a flat boost to downstream task learning, it cannot solve the exploration problem in itself. In Fig. 7, we compare the contribution of `AP` and `CASM` to the aggregated performance of `NGU` and `RECODE` on `DM-HARD-8`. The results consistently demonstrate that `CASM` is a superior representation to `AP` in this domain, leading to significant performance gains with both exploration methods. However, `RECODE` outperforms `NGU` for both representations, indicating that leveraging the representational power for both short-term and long-term novelty signals is a key benefit of our proposed method.

## 6 CONCLUSION

In this paper we introduce `RECODE`, a principled yet conceptually simple exploration bonus for deep RL agents that allows to perform robust exploration by estimating visitation counts from a slot-based memory. `RECODE` improves over prior non-parametric exploration methods by increasing the effective memory span by several orders of magnitude using an online clustering mechanism. Our method sets a new state-of-the-art in task performance on two established exploration benchmarks, Atari's hard exploration subset and `DM-HARD-8`. It is also the first agent to reach the end screen in *Pitfall!* within the time limit which exemplifies `RECODE`'s efficiency of leveraging both long-term (i.e. previous experience) and short-term (i.e. within an episode) novelty signals. Beyond the benchmarks, `RECODE`'s performance also remains unaffected by noisy observations – an adversarial condition which significantly degrades prior approaches such as `RND` and `NGU`. Additionally, we show that our method is agnostic to the concrete representation technique chosen for embedding the observations and scales well with increasingly powerful representations, e.g. using multi-step sequence prediction transformers like our proposed `CASM` architecture. However, `RECODE` is still limited by the choice of the representation and cannot by itself overcome deficiencies stemming from an inappropriate state representation. We also acknowledge that the controllability prior chosen for `CASM` is a strong assumption suitable for the video game environments we experimented with, but this might need to be revisited when `RECODE` is deployed in more realistic, real-world domains. Further details on those limitations are provided in Appendix B. In conclusion, we believe that `RECODE` can serve as a simple yet robust drop-in exploration method compatible with any RL agent and representation learning method which directly translates improvements in representation learning to improvements in exploration performance.

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

## A    RELATED WORKS

In this section, we give a brief and non-exhaustive overview of past works computing visitation counts or estimating densities in RL. We classify them as either parametric or non-parametric.

**Parametric methods.**    Bellemare et al. (2016) and Ostrovski et al. (2017) propose to compute pseudo-visitation counts using density estimators on images such as Context Tree Switching (CTS,Bellemare et al., 2014) or PixelCNN (Van den Oord et al., 2016). On the other hand, Tang et al. (2017) use locality-sensitive hashing to map continuous states to discrete embeddings, where explicit visitation counts are computed. Some methods such as `RND` (Burda et al., 2019) can be interpreted as estimating implicitly the density of observations by training a neural network to predict the output of a randomly initialized and untrained neural network which operates on the observations. Hazan et al. (2019); Pong et al. (2019); Lee et al. (2019); Guo et al. (2021) propose algorithms that search a policy maximizing the entropy of its induced state-space distribution. In particular, the loss optimized by Guo et al. (2021) allows to compute a density estimate as well as maximizing the entropy. Finally, Domingues et al. (2021b) computes a density estimation on top of learned representations, which are inspired by bonuses used in reward-free finite MDPs.

**Non-parametric methods.**    Non-parametric density estimates that we build on date back to Rosenblatt (1956); Parzen (1962) (Parzen–Rosenblatt window) and are widely used in machine learning as they place very mild assumptions on the data distribution. Non-parametric, kernel-based approaches have been already used in RL and shown to be empirically successful on smaller-scale environments by Kveton & Theocharous (2012) and Barreto et al. (2016) and are theoretically analyzed by Ormoneit & Sen (2002); Pazis & Parr (2013); Domingues et al. (2021a). In `NGU` (Badia et al., 2020b), Agent57 (Badia et al., 2020a) and `MEME` (Kapturowski et al., 2022), a non-parametric approach is used to compute a short term reward at the episodic level. Liu & Abbeel (2021) propose an unsupervised pre-training method for reinforcement learning which explores the environment by maximizing a non-parametric entropy computed in an abstract representation space. The authors show improved performance on transfer in Atari games and continous control tasks. Seo et al. (2021) use random embeddings and a non-parametric approach to estimate the state-visitation entropy, but do not generalize to concurrently learned embeddings. Tao et al. (2020) show that K-NN based exploration can improve exploration and data efficiency in model-based RL. While non-parametric methods are good models for complex data, they come with the challenge of storing and computing densities on the entire data set. We tackle this challenge in Sec. 3 of the main text by proposing a method that estimates visitation counts over a long history of states, allowing our approach to scale to much larger problems than those considered in previous works, and without placing assumptions on the representation, that can be trained concurrently with the exploration process and doesn't need to be fixed a priori.

## B    LIMITATIONS

While Atari possesses several challenging sparse reward tasks, the observations are quite simplistic and there is little variation in the environment between episodes. `DM-HARD-8` has much richer observations and several procedurally generated elements such as colors, object shapes, and initial position and orientation; but the level layouts are essentially static. It would be important to understand what additional challenges might be encountered with more complex procedurally-generated elements, and if different representation learning methods might be needed to obtain good generalization across episodes in this setting.

Also, we are aware that concurrent training of embeddings can place some restriction on the maximum discount $\gamma$ that can be effectively used. While this does not limit our method on commonly used RL benchmarks, where episodes tend to be relatively short (e.g. Atari maxes out at $27,000$ steps for the standard action repeat of $4$), it could potentially be an issue for environments with significantly larger timescales, or if there were many more options for what the agent could try between episodes.

`RECODE` can be thought of as a general solution to the exploration problem in RL by biasing the policy towards uniform coverage of the representation space. If the representation does not allow for aliasing together semantically similar states (e.g. if the representation is tabular), exploration can become intractable in large scale environments, since most observations encountered throughout training are

unique. As such, RECODE does not obviate the need to come up with a meaningful representation for the environment at hand. CASM relies on a controllability prior that we find to be well-suited to the RL environments considered in this work, but the question of determining which priors are useful for more general classes of environments remains largely open.

## C GENERAL NOTATION

We consider the usual Reinforcement Learning setting, where an agent interacts with an environment to maximize the sum of discounted rewards, with discount $\gamma \in [0, 1)$, as in Sutton, 1988. In particular, the environment can be described as a Partially-Observable Markov Decision Process (POMDP) Kaelbling et al. (1998). First, we define a Markov Decision Process (MDP) through a tuple $(\mathcal{S}, \mathcal{A}, T, R)$, where $\mathcal{S}$ is the set of states, $\mathcal{A}$ is the set of possible actions, $T$ a transition function, which maps state-actions to distributions over next states, and $R : \mathcal{S} \times \mathcal{A} \to \mathbb{R}$ is the reward function. In particular, a Markov Decision Process is a discrete-time interaction process McCallum (1995); Hutter (2004); Hutter et al. (2009); Daswani et al. (2013) between an agent and its environment. In a Partially-Observable MDP, the agent does not receive a state from $\mathcal{S}$, but an observation $o \in \Omega$, where $\mathcal{O}$ is the function mapping unobserved states to distributions over observations. An observation $o$ will only contain partial observations about the underlying state $s \in \mathcal{S}$. This function can be combined with an intrinsic reward function $r_i$ to enable the exploratory behavior. The environment responds to an agent's action $a \in \mathcal{A}$ by performing a transition to a state $s' \sim T(\cdot|s, a)$; the agent receives a new observation $o' \sim \Omega(\cdot|s')$ and a reward $r \sim R(s, a)$. At step $t$, we can indicate with $h_t = \{o_0, a_1, o_1, \ldots, a_t, o_t\} \in \mathcal{H}_t$ the history of past observations-actions, where $\mathcal{H}_t = \mathcal{H}_{t-1} \times \mathcal{A} \times \mathcal{O}$, $\mathcal{H}_0 = \mathcal{O}$ and the overall history space is $\mathcal{H} = \bigcup_{t \in \mathbb{N}} \mathcal{H}_t$. We consider policies $\pi : \mathcal{H} \to \Delta_{\mathcal{A}}$, that map a history of past observations-actions to a probability distribution over actions.

## D RECODE FROM A CLUSTERING POINT OF VIEW

The update rules RECODE's memory structure in Algorithm 1 of the main text can be interpreted as an approximate inference scheme in a latent probabilistic clustering model. We explore this connection here as means to better understand and justify the proposed algorithm as a density estimator. The rule has a close connection to the DP-means algorithm of Kulis & Jordan (2011), with two key differences:

- the counts of the cluster-centers are discounted at each step, allowing our approach to deal with the non-stationarity of the data due to changes in the policy and the embedding function, effectively reducing the weight of stale cluster-centers in the memory,

- when creating a new cluster-center, we remove an underpopulated one, so as to keep the size of the memory constant.

The adaptations are necessary to accommodate the additional complexities of our setting, which follows a streaming protocol (i.e. data must be explicitly consumed or stored as it arrives, and data that are not stored cannot be accessed again) and is non-stationary (i.e. data are not assumed to be identically distributed as time advances). The clustering algorithm resulting from these adaptations is shown in Algorithm 2. RECODE implements such an algorithm to update the memory, and it also calculates an intrinsic reward for the observed embedding $e$, as described in Section 3 .

---

**Algorithm 2** A streaming clustering algorithm.

1: **Parameters:**

Number of clusters $|M|$

Number of nearest cluster centres $k$

Discounting of counts at each step $\gamma$

Distance threshold to propose the creation of a new cluster $\kappa$

Probability of accepting the creation of a new cluster $\eta$

2: **State:**

Threshold to create new cluster (i.e. average cluster distance) $d = 0$

Cluster centres $m_l = 0 \quad \forall l \in 1 \dots |M|$

Cluster counts $c_l = 0 \quad \forall l \in 1 \dots |M|$

Indices of $k$-nearest neighbours of point $e$: $\text{Neigh}_k(e)$

3: **Implementation:**

4: **for all** received embedding $e \in \{e_0, e_1, e_2, \dots\}$ **do**

5:     Update average inter-cluster distance $d \leftarrow (1 - \tau)d + \frac{\tau}{k} \sum_{l \in \text{Neigh}_k(e)} \|m_l - e\|_2^2$

6:     Discount all cluster-center counts $c_l \leftarrow \gamma \, c_l \quad \forall l \in 1, \dots, |M|$

7:     Find index of nearest cluster center $m_\star = \arg\min_{m \in M} \|m_l - e\|_2$

8:     **if** $\|m_i - e\|_2^2 > \kappa \, d$ and with probability $\eta$ **then**

9:         Sample index $j$ of cluster center to remove with probability $P(j) \propto 1/c_j^2$

10:        Find index of nearest cluster center to $m_j$: $m_\dagger = \arg\min_{m \in M, l \neq j} \|m_l - m_j\|_2$

11:        Redistribute the counts of removed cluster center: $c_\dagger \leftarrow c_j + c_\dagger$

12:        Replace cluster $j$ with a the new cluster at $e$: $m_j \leftarrow e \, , c_j \leftarrow 1$

13:     **else**

14:        Update nearest cluster center $m_\star \leftarrow \frac{c_\star}{c_\star + 1} \mu_i + \frac{1}{c_\star + 1} e$

15:        Update nearest cluster-center count $c_\star \leftarrow c_\star + 1$

16:     **end if**

17: **end for**

---

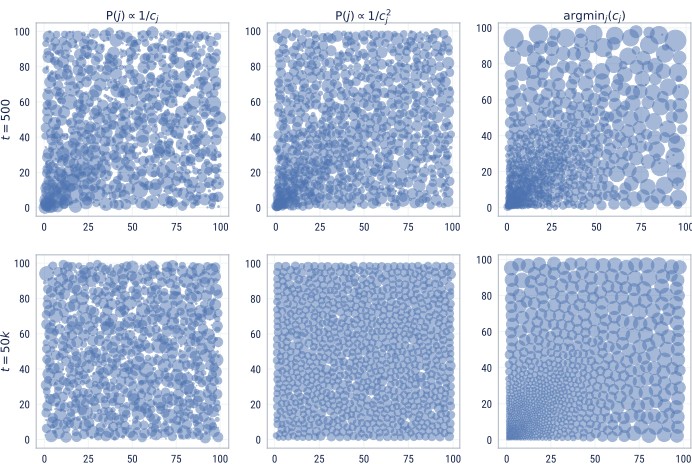

Figure 8: Effect of removal strategy on evolution of cluster centers and counts (with counts corresponding to the size of the marker). At each timestep $t$ we sample a batch of 64 2D-embeddings from a square of side $\min(100, t)$. After $t = 100$ the distribution remains stationary and we would like the distribution of cluster centers and counts to be to become approximately uniform after enough time has passed. For a deterministic removal strategy which selects the clusters with the lowest counts, the cluster centers can remain skewed long after the distribution has stopped changing. For both probabilistic removal strategies, the cluster centers become approximately uniform, but only for the $1/c_j^2$ removal strategy we observe that both cluster centers and counts become uniform. Note that we use a discount of $\gamma = 0.9999$.

### D.1 ADDRESSING FINITE-MEMORY LIMITATIONS.

We first address the modifications introduced to deal with the memory limitations of the streaming setting: 1) each datum (embedding $e_t$ in our notation) is incorporated into a cluster distribution approximation once, then discarded; 2) the total number of clusters is stochastically projected down onto an upper limit on the number of clusters (otherwise they would grow without bound–albeit progressively more slowly). Both modifications allow our method to maintain constant space complexity in the face of an infinite stream of data.

The *step-wise* justification of the Algorithm 2 is relative straightforward. At step $t$, for embedding $e_t$, we show that the following objective is minimised:

$$\min_{l \in 1,...,|M|} \|m_l - e_t\|_2^2 \tag{6}$$
$$\text{s.t.} \quad \|m_l - e_t\|_2^2 \leq \kappa d$$

Working backwards: updating the cluster center reduces the objective directly and will not violate the constraint (unless it was already in violation; this excluded in the precondition of this branch). This accounts for the "else" branch. The "if" branch introduces a new cluster center precisely at $e_t$, thus equation 6 is minimised completely: it is zero for this branch. Finally, selecting the index of the nearest cluster center directly minimises placement of the branch according to equation 6, ignoring the constraint (which is latest ensured by the "if/else"). Note that the hard constraint of equation 6 takes the place of the soft cluster penalty of DP-means (Kulis & Jordan, 2011).

The updates to the cluster centers, unlike k-means and DP-means, are done in an exponentially-weighted moving average of the embeddings, rather than as global optimisation step utilising all of the data. Consequently, and importantly, what happens to equation 6 evaluated for $e_s$, where $s \neq t$, is of significant interest, as objectives for k-means and DP-means account for all data, rather than a single datum.

We tested the qualitative behavior of different removal strategies in Fig. 8. This study suggested that a stochastic removal of a cluster with probability $\propto c^{-2}$ was more stable and better tracked a non-stationary distribution. The intuition we got from these toy examples is also confirmed in ablation experiments ran on the Atari environment, as shown in Fig. 9, where we compare `RECODE` runs with different removal rules.

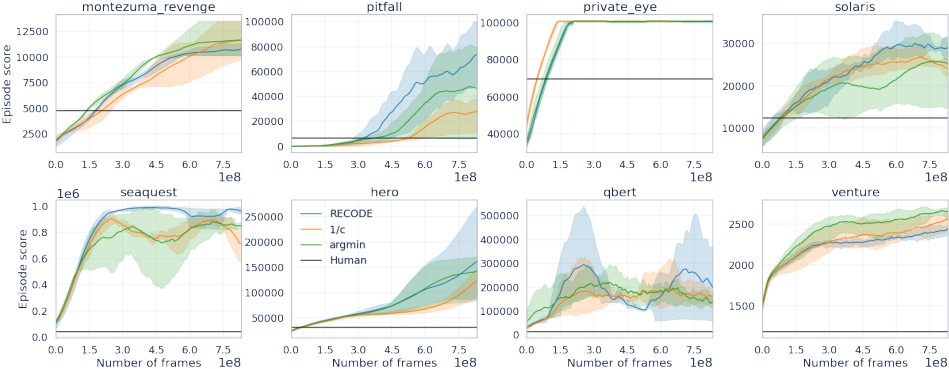

Figure 9: Effect of removal strategy on performance. All choices of removal strategy considered result in a viable algorithm, but there are some environments (most notably *Pitfall!*) where the chosen strategy of $1/c^2$ appears to be more robust.

### D.2 DEALING WITH NON-STATIONARY EMBEDDING DISTRIBUTIONS.

We now turn to the question of how to deal with the non-stationarity of the embedding distribution. We introduced the following modifications to deal with the non-stationarity of our data stream:

1. the cluster count decays,
2. two clusters can be merged to accommodate a new one,

    3. the use of an exponentially weighted moving average update of cluster centers.

In k-means, all of the data are retained. This makes k-means costly: at each step of fitting the entire data set is examined to update the cluster assignments and update the cluster means. Instead, we take a distributional approximation to the data associated with each cluster, and when re-adjusting cluster assignments according to equation 6, we do so in terms of this distributional approximation.

In particular, each cluster is approximated by a Gaussian distribution with precision 1 and whose mean is unknown but with prior zero and precision 1. Specifically:

$$\mu_l \sim \mathcal{N}(0, c_0), \qquad\qquad\qquad e_i|\mu_l \sim \mathcal{N}(\mu_l, 1)$$

where $\mathcal{N}(\mu, \tau)$ denotes a Gaussian (or normal) distribution with mean $\mu$ and precision $\tau$ (precision is the inverse variance). Since the prior on $\mu_l$ is conjugate to the likelihood on $e_i$, we know that the posterior on $m_l$ will have the form $\mathcal{N}(\mu_l, c_l)$. Updating this posterior with a single embedding $e_i$ has the form:

$$m \leftarrow \frac{c_l}{c_l + 1} m + \frac{1}{c_l + 1} e_i, \qquad\qquad c_l \leftarrow c_l + 1$$

This is precisely the update in Algorithm 2.

Note that in this model, the counts $c_l$ are also the precision parameters of the distribution, representing the inverse spread (or the concentration) of each cluster. At each step of Algorithm 2, these counts are decayed. Effectively, this causes the variance of the distribution representing each cluster to spread out: thus at each time step, each cluster becomes less concentrated and more uncertain about which data points belong to it. The hyperparameter $\gamma$ captures the rate of diffusion of all clusters in this manner. This uncertainty increase applied at each step acts as a "forgetting" mechanism that helps the algorithm to deal with a changing data distribution.

Cluster re-sampling, as already justified for $e_t$ above in terms of equation 6, ensures that the number of clusters is bounded by $|M|$. There are two details to examine: what is merged, and how it is merged. As $c_j \mapsto 0$, the probability assigned by the Gaussian likelihood of cluster $j$ to any new datum approaches zero also, thus the cluster with the lowest counts is likely to have the least impact on future density estimates (as it is most diffuse). When $c_j \gg 0$, however, it is not so clear which cluster should be removed. Therefore, we stochastically select which cluster to remove with probability inversely proportional to the square of the counts (using the square of the counts emphasizes small differences in counts more than $1/c_j$). The cluster could potentially be removed completely, but we instead choose to re-assign its counts to the nearest cluster as we experimentally found this strategy to be less sensitive to the choice of hyperparameters.

To help build some intuition about the effects of the discount factor, we illustrate its effects on a toy example with a non-stationary embedding distribution in Fig. 10. We find that tuning the discount $\gamma$ allows to smoothly interpolate between short-term and long-term memory.

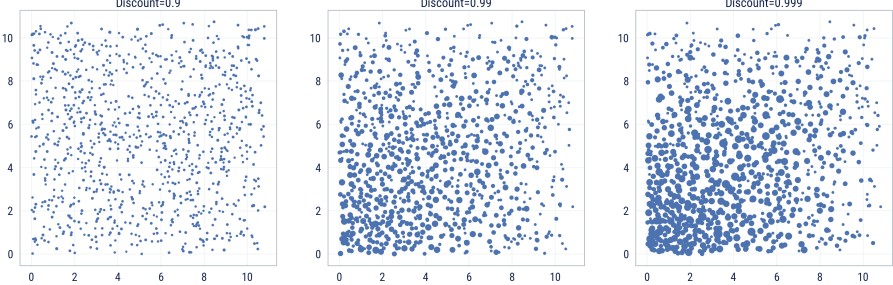

Figure 10: Non-stationary density estimation using `RECODE` on a toy example. For step $t = 0, \ldots, 100$, we sample a batch of $64$ 2D-embeddings uniformly from the square of side $1 + \sqrt{t}$. The support of the embedding distribution therefore expands over time to simulate a non-stationary distribution similar to the distribution of states visited by an RL agent over the course of exploration. We plot the atoms learned by `RECODE` with a size proportional to their count. We find that for a small enough discount, `RECODE` exhibits a short-term memory, accurately approximating the distribution of the final distribution. As we increase the discount, `RECODE` exhibits a longer-term memory, approximating the historical density of states, as can be seen by the concentration of probability mass in the bottom-left corner.

To confirm the practical necessity of cluster discount, we perform additional ablations on the Atari environment. If we don't train the representation during the exploration, but start with a pretrained one, we see that `RECODE` can perform reasonably well also without using discount (see Fig. 12). However, as shown in Fig. 11, as soon as we also train the representation at the same time, the ability to forget old observations allows to compensate distribution-shift and achieves quite better performance.

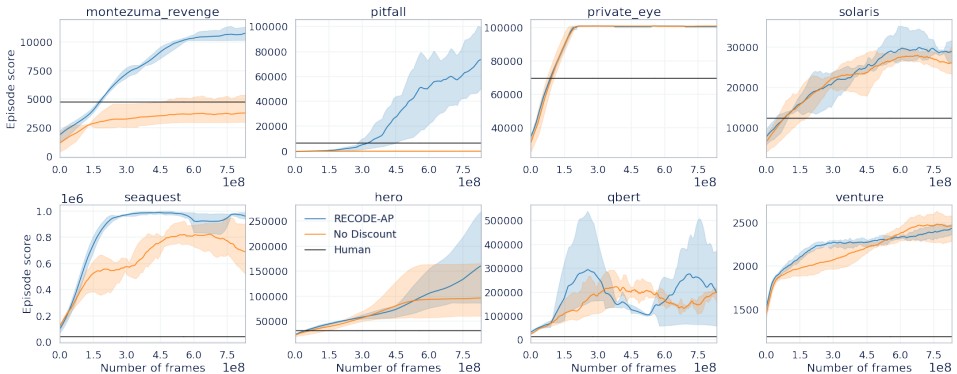

Figure 11: Effect of discount on performance. As embeddings evolve throughout the training it may happen that older clusters stop being meaningful under the current representation. Deactivating the discount (i.e. $\gamma = 1$) results in a significant degradation in performance, especially in hard-exploration settings like *Montezuma's Revenge* and *Pitfall!*.

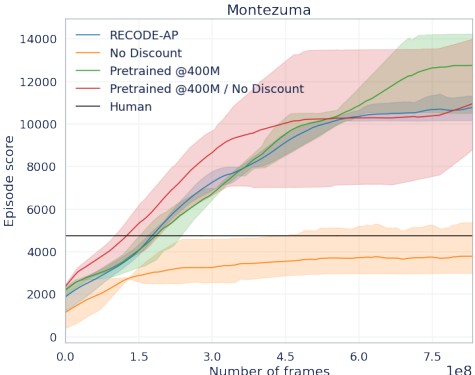

Figure 12: Pretrained vs concurrently-trained embeddings and sensitivity to discount. We take a snapshot of the embeddings at 400M frames and then the agent is trained again from scratch with these frozen embeddings. We find that we can achieve similar performance even with a $\gamma = 1$ (i.e. no discount). Interestingly, we also observe that the agent can achieve much higher scores than those the original agent had achieved at the time the snapshot was taken.

## E ANALYZING EXPLORATION WITH `RECODE`

In this section, we present a simple example to show a simple example to illustrate how the exploration process unfolds for `RECODE`. We use a variant of the *Random Disco Maze*: a grid-world environment proposed in Badia et al., 2020b to show the the importance of estimating the exploration bonus using a controllable state representation, depicted in Fig. 13 (left). The agent starts each episode in a fixed position of a fully observable maze of size 21x21. The agent can take four actions {left, right, up, down}. The episode ends if the agent steps into a wall, reaches the goal state or reaches a maximum of 500 steps.

Crucially the environment presents random variation in the color of each wall fragment at every time step. Specifically, the color of each wall fragment is randomly and independently selected from a set

of five possible colors. This introduces a great deal of irrelevant variability into the system, which presents a serious challenge to exploration bonus methods based on novelty. The reason for this is that the agent will never see the same exact state twice, as the colors of the wall fragments will always be different each time step.

We ran RECODE for 100 million steps. The agent is able to find the goal after collecting around 50 to 60 million steps. In Fig. 13 (right) shows how the distribution of clusters changes as the agent explore this environment. As the agent always starts in the same position (bottom-left corner of the maze), the distribution is heavily skewed towards over-representing this points. We can see that as time progresses the cluster centers uniformly cover all the maze.

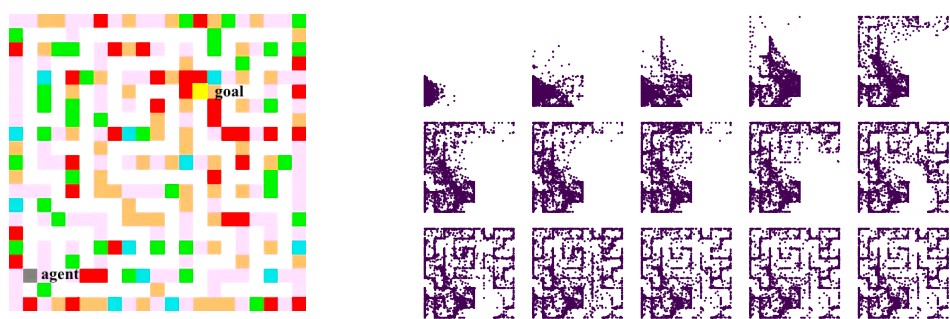

Figure 13: (Left) Random Disco Maze (Right) Evolution of the distribution of the clusters learned by RECODE over time, see text for details.

We investigate how far back the memory of RECODE goes in *Montezuma's Revenge*. The results are shown in Fig. 14. We find that the distribution of the age of the clusters learned by RECODE (i.e. how many steps ago each atom has been inserted in the memory) exhibits a mode around $2 \cdot 10^6$ actor steps, which corresponds to hundreds of episodes, with a significant number of clusters ten times older than that. We remind that NGU's short-term non-parametric novelty estimated at most one episode (red line in the Figure).

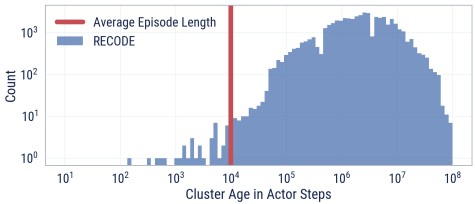

Figure 14: Age distribution of the clusters learned by RECODE on *Montezuma's Revenge*. RECODE's memory horizon spans much more than a single episode. We set $\gamma = 0.999$ as in the experiments of Fig. 4. We indicate in red the average length of an episode, showing that in this setting, RECODE's memory reaches back thousands of episodes.

Table 1: Atari Hyper-parameters.

| Parameter | Value |
|---|---|
| RECODE memory size | $5 \times 10^4$ |
| RECODE discount $\gamma$ | 0.999 |
| RECODE insertion probability $\eta$ | 0.05 |
| RECODE relative tolerance $\kappa$ | 0.2 |
| RECODE reward constant $c$ | 0.01 |
| RECODE decay rate $\tau$ | 0.9999 |
| RECODE neighbors $k$ | 20 |
| IM Reward Scale $\beta_{\mathrm{IM}}$ | 1.0 |
| Max Discount | 0.9997 |
| Min Discount | 0.97 |
| Replay Period | 80 |
| Trace Length | 160 |
| Replay Ratio | 6.0 |
| Replay Capacity | $2 \times 10^5$ trajectories |
| Batch Size | 64 |
| RL Adam Learning Rate | $3 \times 10^{-4}$ |
| Emedding Adam Learning Rate | $6 \times 10^{-4}$ |
| RL Weight Decay | 0.05 |
| Embedding Weight Decay | 0.05 |
| RL Torso initial stride | 4 |
| RL Torso num blocks | $(2, 3, 4, 4)$ |
| RL Torso num channels | $(64, 128, 128, 64)$ |
| RL Torso strides | $(1, 2, 2, 2)$ |

## F  HYPER-PARAMETERS AND COMPUTATIONAL RESOURCES

We implemented RECODE and all the baseline agents and novelty mechanisms in a distributed setting (see App. G and H). We report here the total computational infrastructures used by each distributed agent, (including multiple actors, learner and RECODE memory mechanism where applicable). One seed for an Atari experiments (e.g., for MEME-RECODE-AP and MEME-NGU-AP) took 24h to execute using multiple servers with a total of 64 CPUs, 1TB RAM, and 5 TPUv4. One seed for a DM-HARD-8 experiment (e.g., for MEME-RECODE-CASM and MEME-NGU-CASM) took 90h to execute using multiple servers with a total of 512 CPUs, 1TB RAM, and 5 TPUv4.

We also report here the precise hyper-parameter values used in our experiment, Table 1 for Atari and Table 2 for DM-HARD-8 We omit hypers which do not differ from the base MEME agent Kapturowski et al. (2022).

We emphasize that the relevant hyperparameters for RECODE, i.e. those for which the algorithm is sensitive to changes, are only

- RECODE memory size

- RECODE discount $\gamma$

See the previous Appendix D and the main text for the discussion of their interpretation. The other parameters are particular choices, for which the algorithm proved to be robust in many different environments, and we did not need to retune them. We ran additional experiments sweeping across all permutations of $\eta \in 0.05, 0.2$ and memory size $\in [5 \times 10^4, 2 \times 10^5]$ in both Atari and DM-HARD-8 and found that performance in most environments had little sensitivity to these choices.

Table 2: `DM-HARD-8` Hyper-parameters.

| Parameter | Value |
|---|---|
| `RECODE` memory size | $2 \times 10^5$ |
| `RECODE` discount $\gamma$ | 0.997 |
| `RECODE` insertion probability $\eta$ | 0.2 |
| `RECODE` relative tolerance $\kappa$ | 0.2 |
| `RECODE` reward constant $c$ | 0.01 |
| `RECODE` decay rate $\tau$ | 0.9999 |
| `RECODE` neighbors $k$ | 20 |
| IM Reward Scale $\beta_{\text{IM}}$ | 0.1 |
| Max Discount | 0.997 |
| Min Discount | 0.97 |
| Replay Period | 40 |
| Trace Length | 80 |
| Replay Ratio | 2.0 |
| Replay Capacity | 5000 trajectories |
| Batch Size | 128 |
| RL Adam Learning Rate | $1 \times 10^{-4}$ |
| Embedding Adam Learning Rate | $3 \times 10^{-4}$ |
| RL Weight Decay | 0.1 |
| Embedding Weight Decay | 0.1 |
| RL Torso initial stride | 2 |
| RL Torso num blocks | $(2, 4, 12, 6)$ |
| RL Torso num channels | $(64, 128, 128, 64)$ |
| RL Torso strides | $(1, 2, 2, 2)$ |

Table 3: `CASM` Hyper-parameters.

| Parameter | Value |
|---|---|
| Transformer Type | GatedTransformerXL |
| State Mask Rate | 0.8 |
| Num Masks Per Trajectory | 4 |
| Action Embedding Size | 32 |
| Num Layers | 2 |
| Attention Size | 128 |
| Num Attention Heads | 4 |
| MLP Hidden Sizes | $(512,)$ |
| Predictor Hidden Sizes | $(128,)$ |

# G ARCHITECTURE OF A DISTRIBUTED AGENT USING `RECODE`

We now detail how `RECODE` can be efficiently integrated in a typical distributed RL agent (Espeholt et al., 2018; Kapturowski et al., 2018) that comprises several processes that run in parallel and interact with each other, allowing for large-scale experiments. Classically, a Learner performs gradient steps to train a policy $\pi_\theta$ and an embedding (representation) function $f_\theta$, forwarding the parameters $\theta$ to an Inference Worker. A collection of independent Actors query the inference worker for actions that they execute in the environment and send the resulting transitions to the Learner, optionally through a (prioritized) Replay (Mnih et al., 2015; Schaul et al., 2015). When using `RECODE`, the Actors additionally communicate with a shared Memory implementing Algorithm 1: at each step $t$, they query from the Inference Server an embedding $f_\theta(h_t)$ of their history and send it to the shared Memory which returns an intrinsic reward $r_t$ that is then added to the extrinsic reward to train the policy in the Learner process. A diagram giving an overview of the typical architecture of a distributed agent using `RECODE` is given in Figure 15.

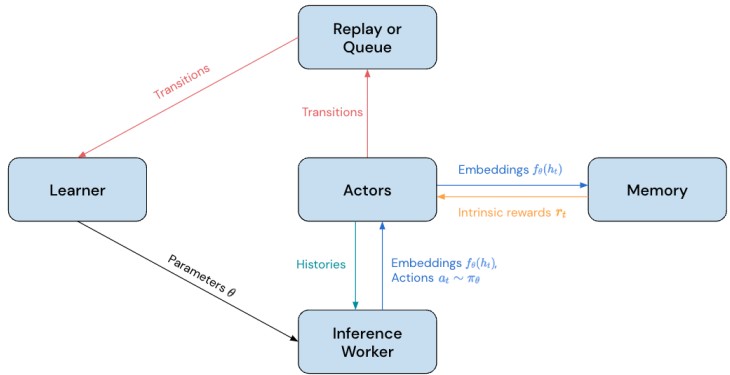

Figure 15: Overview of the architecture of a distributed agent using `RECODE`.

## H  AGENT TAXONOMY

All methods evaluated in the experiments in the main paper and the extended experiments in App. L are composed by three main components.

- A base agent that oversees the overall RL learning process (e.g., executing actions and collection observations, computing adjusted returns, updating the policy, ...). We focus on `MEME` (Kapturowski et al., 2022), a recent improvement over Agent57 (Badia et al., 2020a) that achieves much greater sample efficiency and is the current state-of-the-art on Atari, and a VMPO-based agent (Guo et al., 2022) that is the current state-of-the-art on `DM-HARD-8`.

- A representation learning mechanism to generate observation embeddings which are fed to the intrinsic reward generator. We consider both Action Prediction (`AP`) and `CASM` embeddings. Note that some intrinsic reward modules cannot make effective use of the representation learning module (e.g., `RND`), while others merge both second and third module in a single approach (e.g., `BYOL-Explore`)

- An algorithm to generate intrinsic rewards. In addition to `RECODE`, we also consider the recent `BYOL-Explore` Guo et al. (2022), `NGU` Badia et al. (2020b) and `NGU`'s two building blocks, `RND` Burda et al. (2019) and Episodic Memory (`EMM`) Pritzel et al. (2017).

For example, in our more detailed taxonomy the original `MEME` agent described in Kapturowski et al. (2022) is denoted as the `MEME-NGU-AP` baseline, and compared against our novel `MEME-RECODE-AP` agent where the only modifications is the changed exploration reward. Table L reports more details on all combinations available present in our experiments.

Table 4: Taxonomy of agents used in the experiments.

| Agent name | | Base agent | Intrinsic reward | Representation learning |
|---|---|---|---|---|
| `MEME-NGU-AP` | Kapturowski et al. (2022) | `MEME` | `NGU` | `AP` |
| `MEME-RND` | (ablation) | `MEME` | `RND` | N/A[a] |
| `MEME-EMM-AP` | (ablation) | `MEME` | `EMM` | `AP` |
| `MEME-RNDonAP` | (ablation) | `MEME` | `RND` | `AP` [b] |
| `MEME-RECODE-AP` | (this paper) | `MEME` | `RECODE` | `AP` |
| `MEME-RECODE-CASM` | (this paper) | `MEME` | `RECODE` | `CASM` |
| `MEME-NGU-CASM` | (ablation) | `MEME` | `NGU` | `CASM` |
| `VMPO-BYOL-Explore` | Guo et al. (2022) | `VMPO` | `BYOL-Explore` | `BYOL-Explore` [c] |

(a) As in the original paper `RND` takes as input raw observations.

(b) To test `RND`'s ability to cope with non-stationary representations, we train an `AP` encoder concurrently with the policy and use it to create embeddings of the observations that are fed in `RND` (i.e., running `RND` on top of `AP`).

(c) The `BYOL-Explore` mechanism internally trains a neural network to predict the dynamical evolution of the observations. This provides the agent with both a reward/novelty signal (prediction error) as well as an embedded representation of the observations (that can be extracted from the last few layers of the network).

# I EXPLORATION IN THE DM-HARD-8 ENVIRONMENT

The three-dimensional tasks in DM-HARD-8 can have an extremely large state space to explore. Consider for example the Baseball task, as shown in Fig. 16: the agent needs to look at the scene, find the bat, pick it up, throw the ball down, pick up the ball and be able to get the apple.

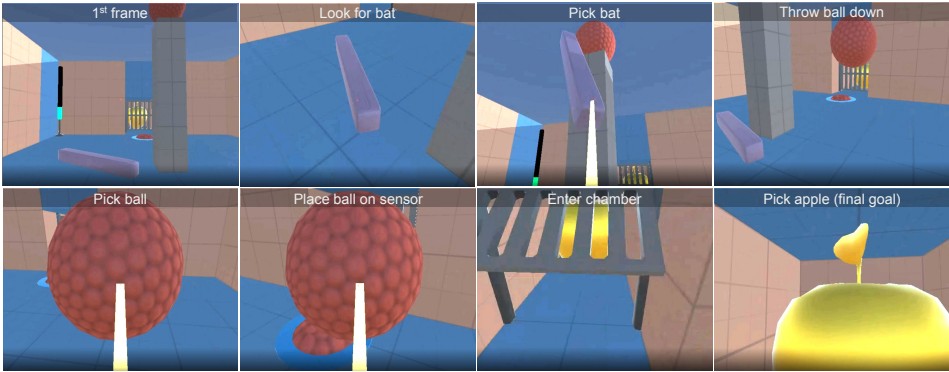

Figure 16: First-person-view snapshots of an agent solving the DM-HARD-8 Baseball task. Images are ordered chronologically from left to right and top to bottom. Each image depicts a specific stage of the task. The agent must interact with specific objects in the environment in order to solve the task.

# J AGGREGATED RESULTS

In this section, we show the aggregated results over all different environments, both for the Atari suite and for DM-HARD-8. To ensure that no single environment dominates due to larger reward scales we use the Human Normalized Score (Mnih et al., 2015) in each environment, and then cap scores above 100% prior to averaging.

As Fig. 17 shows, (Left), the uncapped score can swing significantly over time, which in this case is simply an artifact the high variance present in *Q\*bert*. This variance arises due to a bug in *Q\*bert*, which allows for much larger scores to be obtained if exploited.

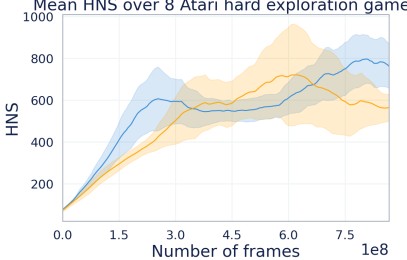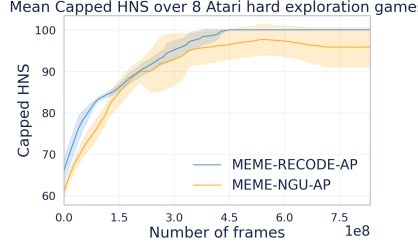

Figure 17: Aggregated results on Atari. (Left): Mean Human-Normalized Scores of MEME-RECODE-AP compared to MEME-NGU-AP on Atari games. (Right): Capped Human-Normalized Scores.

In Fig 18, we present our main results aggregated over all environments in each DM-HARD-8 task suite. Table 6 summarizes the results and compares the performance of the RECODE novelty reward mechanism with that of NGU. We emphasize how effective our approach is when applied to three-dimensional environments like DM-HARD-8, if compared to alternatives like BYOL-Explore.

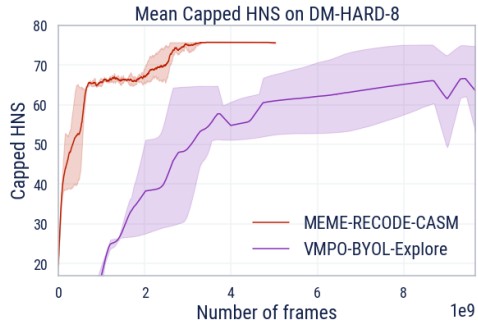

Figure 18: Aggregated results on `DM-HARD-8`. Mean Capped Human-Normalized Scores of `MEME-RECODE-CASM` are compared to `VMPO-BYOL-Explore`.

Table 5: Aggregated results over `DM-HARD-8` environment tasks.

| Game | Human | MEME-NGU-AP | MEME-NGU-CASM | MEME-RECODE-AP | MEME-RECODE-CASM |
|---|---|---|---|---|---|
| Baseball | 7.90 | $0.00 \pm 0.00$ | $0.00 \pm 0.00$ | $\mathbf{10.00 \pm 0.00}$ | $\mathbf{10.00 \pm 0.00}$ |
| Drawbridge | 12.30 | $0.00 \pm 0.00$ | $4.15 \pm 6.62$ | $12.41 \pm 0.07$ | $\mathbf{12.86 \pm 0.13}$ |
| Navigate Cubes | 7.80 | $3.33 \pm 5.33$ | $9.76 \pm 0.25$ | $\mathbf{10.00 \pm 0.00}$ | $\mathbf{10.00 \pm 0.00}$ |
| Push Blocks | $\mathbf{8.40}$ | $0.00 \pm 0.00$ | $0.24 \pm 0.24$ | $1.07 \pm 0.62$ | $4.06 \pm 2.14$ |
| Remember Sensor | $\mathbf{7.60}$ | $0.00 \pm 0.00$ | $0.00 \pm 0.00$ | $0.00 \pm 0.00$ | $0.00 \pm 0.00$ |
| Throw Across | 5.70 | $0.00 \pm 0.00$ | $0.00 \pm 0.00$ | $0.00 \pm 0.00$ | $\mathbf{9.72 \pm 0.31}$ |
| Wall Sensor | 9.10 | $0.00 \pm 0.00$ | $0.00 \pm 0.00$ | $\mathbf{10.00 \pm 0.00}$ | $\mathbf{10.00 \pm 0.00}$ |
| Wall Sensor Stack | $\mathbf{8.60}$ | $0.00 \pm 0.00$ | $0.00 \pm 0.00$ | $0.00 \pm 0.00$ | $0.00 \pm 0.00$ |

Table 6: Aggregated results over Atari environment tasks.

| Game | Human | MEME-NGU-AP | MEME-RECODE-AP | p-value |
|---|---|---|---|---|
| | | Final Performance | | |
| Montezuma's Revenge | 4753.3 | $\mathbf{10715.2 \pm 4967.4}$ | $\mathbf{11591.94 \pm 1112.67}$ | 0.1970 |
| Pitfall! | 6463.7 | $44947.6 \pm 13020.0$ | $\mathbf{77737.38 \pm 14669.51}$ | $\mathbf{0.0043}$ |
| Private Eye | 69571.3 | $\mathbf{100796.2 \pm 2.5}$ | $100794.7 \pm 2.0$ | 0.3496 |
| Solaris | 12326.7 | $19810.1 \pm 5060.0$ | $\mathbf{26152.5 \pm 4503.0}$ | 0.1970 |
| Seaquest | 42054.7 | $\mathbf{782330.1 \pm 159871.7}$ | $\mathbf{793697.7 \pm 184030.5}$ | 0.4091 |
| Hero | 30826.4 | $\mathbf{187244.3 \pm 27855.2}$ | $141638.4 \pm 60803.2$ | $\mathbf{0.0465}$ |
| Q*Bert | 13455 | $57751.0 \pm 26942.8$ | $\mathbf{182638.3 \pm 98698.3}$ | $\mathbf{0.0465}$ |
| Venture | 1187.5 | $\mathbf{2552.2 \pm 96.1}$ | $2502.4 \pm 78.7$ | 0.3496 |
| | | AUC | | |
| Montezuma's Revenge | | $\mathbf{6742.1 \pm 2144.8}$ | $\mathbf{7498.1 \pm 130.8}$ | 0.1970 |
| Pitfall! | | $\mathbf{25784.5 \pm 12320.7}$ | $\mathbf{26775.0 \pm 6949.1}$ | 0.3496 |
| Private Eye | | $\mathbf{86109.6 \pm 3649.4}$ | $\mathbf{89980.2 \pm 1377.6}$ | 0.1548 |
| Solaris | | $15338.7 \pm 2569.7$ | $\mathbf{22454.0 \pm 600.4}$ | $\mathbf{0.0011}$ |
| Seaquest | | $631153.4 \pm 67020.8$ | $\mathbf{730154.4 \pm 86270.1}$ | $\mathbf{0.0325}$ |
| Hero | | $\mathbf{89115.6 \pm 8964.1}$ | $71842.1 \pm 16767.4$ | 0.0898 |
| Q*Bert | | $\mathbf{148095.1 \pm 67841.9}$ | $\mathbf{166216.5 \pm 13799.1}$ | 0.0898 |
| Venture | | $\mathbf{2322.5 \pm 27.3}$ | $2253.8 \pm 61.6$ | 0.1201 |
| Mean HNS | | $543.6_{(477.9, 600.8)}$ | $\mathbf{715.5_{(606.1, 833.9)}}$ | $\mathbf{0.0185}$ |
| Median HNS | | $\mathbf{329.4_{(234.9, 430.9)}}$ | $\mathbf{357.9_{(270.1, 473.6)}}$ | 0.3551 |
| Mean Capped HNS | | $95.7_{(89.4, 100.0)}$ | $\mathbf{100_{(100, 100)}}$ | $\mathbf{0.0419}$ |

Table 7: Atari final performance and AUC. For individual games we use a one-sided Mann-Whitney U test for difference in mean between `RECODE` and `NGU` and report the corresponding p-values. For aggregate statistics (Mean and Median HNS) we compute p-values using a bootstrap estimate.

# K  MULTITASK EXPERIMENTS

We also implemented `RECODE` in a VMPO-based agent similar to the one used with BYOL-Explore (Guo et al., 2022), and compared our performance with BYOL-Explore in the multi-task setting. This experiment serves two different purposes. First, this demonstrates the generality of our exploration bonus, that is shown to be useful in widely different RL agents, be they value-based or policy-based. Second, we can do a direct comparison with the state of the art BYOL-Explore agent in the multi-task settings. However, we note that the representation learning technique used in this experiment, 1-step Action Prediction, is based on a feed-forward embedding that discards past history, and may therefore not be the best fit for exploration in Partially Observable MDPs (POMDPs). Still, Fig. 19 shows that `RECODE`'s performance is competitive with that of BYOL-Explore, with only one level missing to match its performance. Improving this performance using better-suited representations, such as `CASM`, is left for future work.

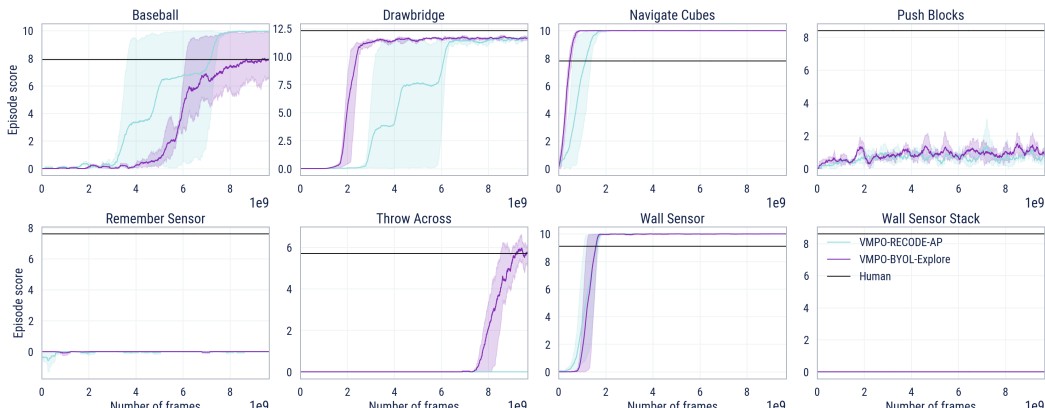

Figure 19: Performance of `RECODE` compared to BYOL-Explore on the multi-task version of `DM-HARD-8`. Our `RECODE` implementation in this experiments is based on VMPO, using a continuous action set.

# L  ADDITIONAL ABLATION STUDIES

## L.1  MEMORY SIZE AND INSERTION PROBABILITY.

We report here additional results on the performance of `RECODE` (in particular a `MEME-RECODE-AP` agent) on Atari for different memory sizes $\{2 \cdot 10^5, 5 \cdot 10^4\}$ and $\eta \in \{0.2, 0.05\}$.

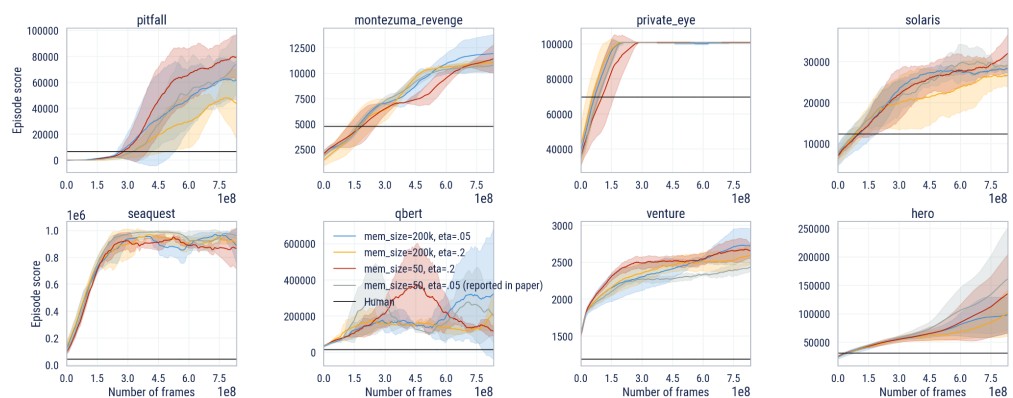

Figure 20: Ablation study on $\eta$ and memory size for `RECODE`. The combination $\eta = 0.05$ and memory size 50k is the one reported in the main paper.

We see that for all combinations `RECODE` achieves a robust performance on most environments, never failing to achieve super-human performance.

## L.2 CASM MASKING

Here we analyze the performance of our technique when removing the masking strategy in `CASM` (see main text). For each element of the sequence, instead of providing either the embedded observation or the action, we always provide both, making the classifier upstream task too simple. Masking allows to provide extra context (with respect to action-prediction) while keeping the prediction task hard enough to require the encoding of high-level features in the representation.

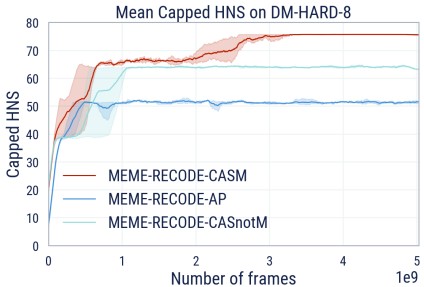 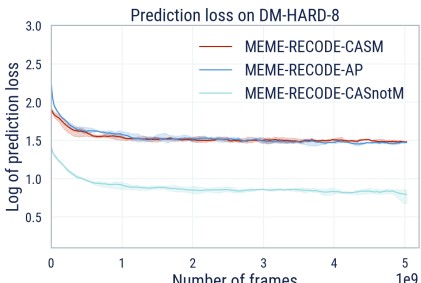

Figure 21: `CASM` masking ablation. (Left): `CASnotM` is `CASM` without any masking applied to the trajectory. Even by itself, the extra context provides a clear advantage over `AP`, but the masking strategy is essential to solve Push Block. (Right): The loss for `CASnotM` is much lower than for both `AP` and `CASM` suggesting that the additional context without any masking makes the prediction task easier (but leading to a less robust representation).

## L.3 RND ON TOP OF ACTION PREDICTION EMBEDDINGS

We adapt `RND` to leverage trained action-prediction embeddings, which we refer to as `RNDonAP`. To that effect, we use a randomly initialized Multi-Layer Perceptron (MLP) to perform a random projection of the embedding, and use a second, trained MLP, to reconstruct this random projection. The reconstruction error provides an intrinsic reward for exploration, which we normalize by a running estimate if its standard deviation as in Burda et al. (2019). We find that the resulting agent is unable to solve some of the hardest exploration games such as `Montezuma's Revenge` or `Pitfall!`. The results of this ablation is shown in Fig. 22. Experiments with pre-trained embeddings do seem to indicate that `RNDonAP` can obtain stronger performance in this setting, but the inability to concurrently train the embeddings greatly limits the general applicability of the method.

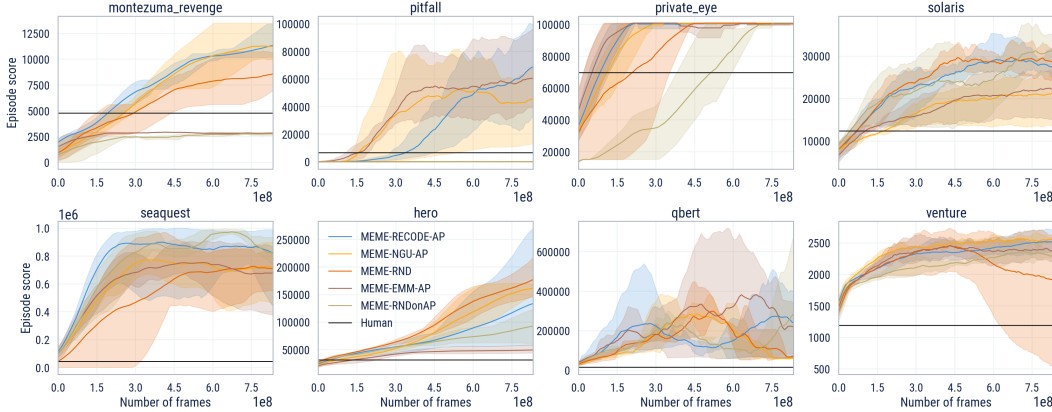

Figure 22: Performance of `RECODE` compared to `MEME` and its ablations on 8 hard exploration Atari games. We find that this approach does not allow to solve some of the hardest games such as *Montezuma's Revenge* or *Pitfall!*

One possible explanation of this failure is the fact that a large `RND` error can be caused by either the observation of a new state, or a drift in the representation of an already observed one. The failure of `RND` to disentangle these two effects results in poor exploration.

## L.4 RECODE vs NGU comparison on pre-trained embeddings.

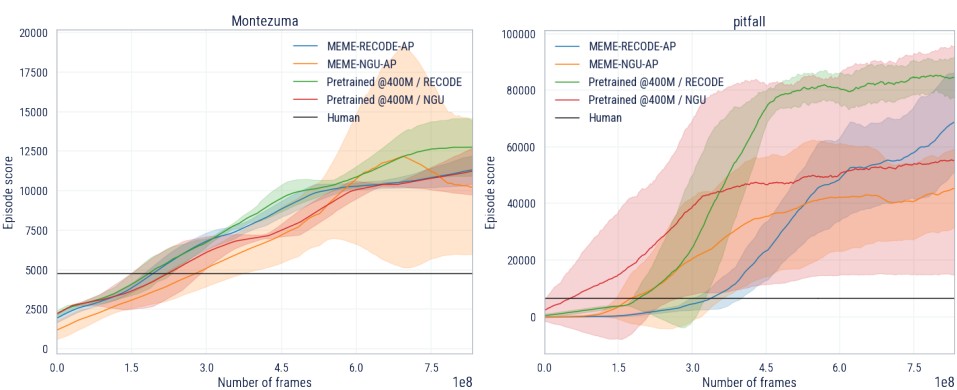

Figure 23: Performance with pretrained embeddings used with or `NGU`.

In this ablation we test how much the performance of `NGU` can be improved when using pre-trained embeddings. As one of the main limitations of the `RND` component in `NGU` is the low compatibility with concurrently-trained embeddings, we expect that when using high quality pre-trained and *fixed* embeddings `RND` (and thus `NGU`) can perform much better.

We evaluate this in the two most representative games from the Atari suite, *Montezuma's revenge* and *Pitfall!*. For each game, we first run `RECODE` with concurrently-trained embeddings, and then take a snapshot of the embeddings at 400M frames. The agent is then trained again from scratch with these frozen embeddings using either `RECODE` (with further concurrent training) or `NGU` (further training only for the `EMM` part). In Figure 23 we report results for the original `RECODE` run, the new runs with pre-trained embeddings as well as a non-pretrained `NGU` run for reference. We find that on *Montezuma's revenge*, where there are less visual confounding factors and a policy learned directly in pixel space is more effective, pre-training brings only small improvements for both `RECODE` and `NGU`. However in *Pitfall!*, where it is important to have a good representation to filter out uncontrollable elements, agents that leverage pre-trained embeddings can achieve much higher scores than those the original agent had achieved at the time the snapshot was taken.

## L.5 CASM in Atari

As `CASM` was specifically designed to aid representation learning in partially observable and 3D environments, it might be expected to be less beneficial in environments such as Atari which have a more limited degree of partial observability. Indeed, performance is quite similar between `MEME-RECODE-CASM` and `MEME-RECODE-AP` across most games, with the notable exception of Hero, in which `CASM` yields a significant performance boost.

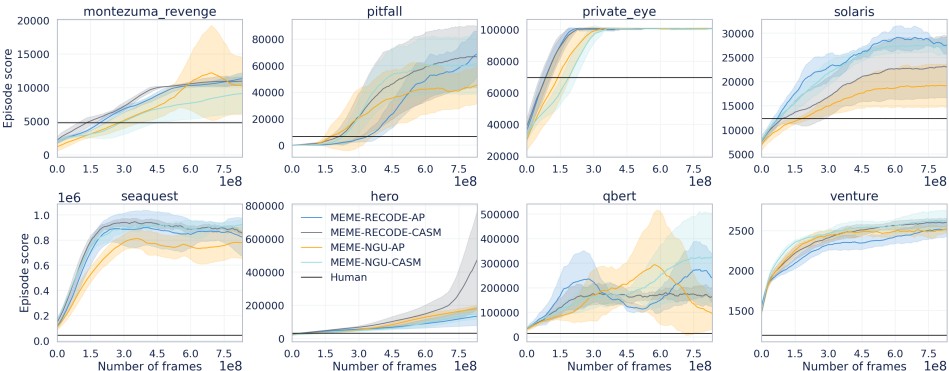

Figure 24: Comparison of `CASM` with AP on Atari. All `CASM` hypers are identical to those used in `DM-HARD-8` except for the state mask rate (set to $0.1$ in these experiments, as we observed that high values exhibited much higher variance between seeds on *Pitfall!*)

## L.6 `RECODE` on top of BYOL embeddings

To further show `RECODE` robustness to change of representation, in Fig. 25, we compare the performance of the `RECODE` embedding on top of an action-prediction representation with respect to a BYOL representation (as in Guo et al. (2022)).

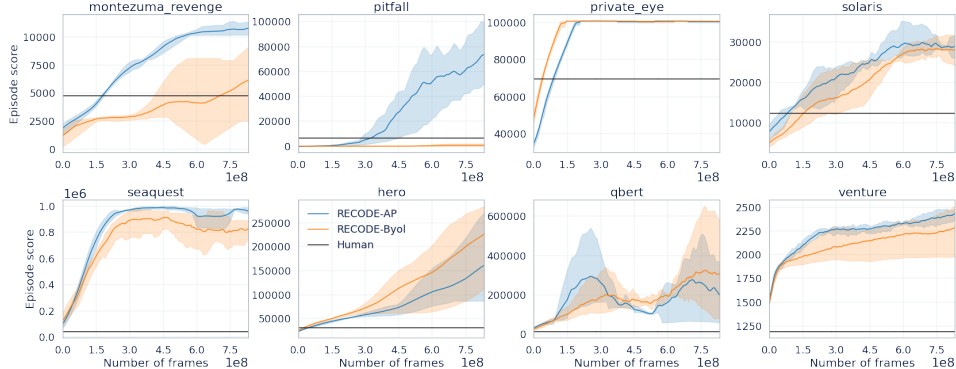

Figure 25: Comparison of `RECODE` with AP vs BYOL embeddings. We observe that `RECODE` is able to leverage BYOL embeddings to achieve superhuman performance on *Montezuma's Revenge* and achieve positive scores on *Pitfall!*, though significantly underperforming compared to AP embeddings. For BYOL, we swept over embedding sizes of $\{32, 128, 512\}$ and report the best performing size, $32$.

