# OpenReview forum: "Unlocking the Power of Representations in Long-term Novelty-based Exploration"
_ICLR.cc/2024/Conference — ICLR 2024 spotlight_

### Official Review · Reviewer_CuGy · 2023-10-12

**Soundness:** 3 good
**Presentation:** 3 good
**Contribution:** 3 good
**Rating:** 8
**Confidence:** 4

**Summary:**

This work presents a framework that integrates sophisticated representation learning with long-term memory to improve novelty-seeking behaviours in deep reinforcement learning. The framework consists of two key components: RECODE, a memory buffer that approximates visitation counts in a learned embedding space; and CASM, a transformer model that learns the embedding space by jointly addressing masked sequence modelling and inverse dynamics prediction.

The paper explores and implements various components to tackle challenges inherent in deep RL, such as non-stationary distributions, online learning of representations, and sparse rewards. Experimental results on challenging benchmark tasks demonstrate that RECODE (the memory buffer) enhances performance regardless of the representation learning method, and CASM improves performance independently of the memory-based approach. The algorithm is compared with NGU and Byol-Explore, two recent and popular deep exploration algorithms.

Overall, the paper presents a sophisticated algorithm that achieves great performance in hard-exploration tasks, and its implementation details are well-justified and clearly presented.

**Strengths:**

The paper builds upon prior work in the fields of exploration in deep reinforcement learning and representation learning. Concerning RL-related work, it extends existing non-parametric novelty estimation methods that maintain a history of observations in a memory buffer and use KNN to compute similarity. However, this work pushes the boundaries of previous memory-based algorithms by operating the memory buffer in a much more compact embedding space. Additionally, the paper presents several implementation details to ensure an accurate estimation of true visitation counts, addressing challenges related to non-stationary data and representation distributions. For representation learning, the paper adopts the widely recognized transformer architecture for sequence modelling and employs masked training to learn compact representations in reinforcement learning. Furthermore, the study incorporates the established inverse dynamics prediction objective but frames it within the more condensed embedding space. This work effectively combines and adapts robust foundations from multiple fields into a high-quality and well-justified framework for reinforcement learning, resulting in outstanding practical performance.

The paper maintains clarity, offering comprehensive descriptions of the background theory and the architecture of the presented framework. It includes numerous helpful figures and algorithm pseudo-code, enhancing the reader's understanding of the method. Notably, the paper meticulously addresses significant challenges related to long-horizon exploration in deep RL, each thoroughly examined with dedicated experiments and figures that surpass typical task-return analyses.

In summary, this work makes a significant contribution to the community. Specifically, it introduces an algorithm that harnesses potent representation learning and precise novelty estimation to solve challenging long-horizon exploration problems effectively.

**Weaknesses:**

The presented algorithm demonstrates impressive performance in well-established hard-exploration benchmarks. However, it lacks evaluation in an open-ended (yet equally challenging in terms of exploration) environment, such as Minecraft, which would combine the complexities of long-horizon tasks from Atari with rich observations from environments like DM-Hard-8, as utilized in the paper.

While the authors assert the robustness of the presented algorithm to different RL algorithms for policy learning, the RL training solely relies on MEME, a powerful RL algorithm with substantial learning capacity. While this choice appears suitable, the paper lacks an analysis showcasing the performance of more commonly used algorithms like PPO or DQN when coupled with RECODE. Including such comparative results would significantly support the paper's claims regarding RECODE's versatility in enhancing long-horizon exploration across various RL algorithms. VMPO is also used for the DM-Hard-8 suite, but I believe this doesn't provide evidence that RECODE would still work well with more popular RL algorithms.

**Questions:**

The computational resource requirements for training RECODE remain unclear. The paper indicates a substantial number of environment interactions, around 7.5e8 steps, for various environments, comparable to previous studies. However, essential details such as memory usage, CPU requirements, and GPU memory are not provided. This lack of information poses a significant concern, as it could either limit the framework's practical applicability or potentially underscore its uniqueness as a noteworthy contribution.

---

> ### Author Response · Authors · 2023-11-22
> **Re: Minecraft experiments**
>
> > The presented algorithm demonstrates impressive performance in well-established hard-exploration benchmarks. However, it lacks evaluation in an open-ended (yet equally challenging in terms of exploration) environment, such as Minecraft, which would combine the complexities of long-horizon tasks from Atari with rich observations from environments like DM-Hard-8, as utilized in the paper.
>
> Minecraft is definitely an interesting environment for exploration that combines complex long-horizon tasks and rich observation environments. However, given the complexity of it, the current SOTA approaches combine additional ingredients to exploration to make progress.
> For example, to make exploration bonuses effective they are combined with video pre-training (https://openai.com/research/vpt, https://minerl.io/) or shaped intrinsic rewards to encourage predetermined policies (e.g. https://openreview.net/pdf?id=ZUXy6d49JNJ). In https://arxiv.org/pdf/2303.16563.pdf a naive state visitation count is implemented in addition to more domain-specific similarity-based rewards based on pre-trained large language models.
>
> In our paper, we propose a novelty-based approach for exploration that does not take into account any domain-specific knowledge about the environment (e.g.hand-engineered action spaces, skills needed for complex tasks, video pretraining). Therefore, we focus on testing its performance in environments where other competing fully-intrinsic approaches can make progress.  In Minecraft, a pure intrinsically motivated agent, with an intrinsic reward that does not take into account specific skills to mimic, would likely take a very long time to discover complex skills (e.g. getting a diamond pickaxe), and it would still be necessary to benchmark the performance of the intrinsic exploration mechanism on simpler tasks.
>
> We would expect that our technique can improve the performance of existing techniques on long-horizon tasks on Minecraft when used as a replacement for naive novelty-based rewards, when including some bias towards useful skills or imitation based pretraining and believe that it is an important direction for future research to investigate how to reduce the need for prior knowledge in discovering such complex skills.

---

> ### Author Response · Authors · 2023-11-22
> **Re: robustness to different RL algorithms**
>
> > While the authors assert the robustness of the presented algorithm to different RL algorithms for policy learning, the RL training solely relies on MEME, a powerful RL algorithm with substantial learning capacity. While this choice appears suitable, the paper lacks an analysis showcasing the performance of more commonly used algorithms like PPO or DQN when coupled with RECODE. Including such comparative results would significantly support the paper's claims regarding RECODE's versatility in enhancing long-horizon exploration across various RL algorithms. VMPO is also used for the DM-Hard-8 suite, but I believe this doesn't provide evidence that RECODE would still work well with more popular RL algorithms.
>
> Indeed, the primary piece of supporting evidence of this claim is the strong performance obtained by VMPO-RECODE in the multitask setting without any additional tuning. We agree that the evidence would be much stronger if a wider variety of popular algorithms were tested. Since implementing and validating performance of any additional algorithms used for such an analysis would require a significant engineering effort, it is not feasible to obtain these results prior to the end of the rebuttal period, but we will look into possibilities to include another off-the-shelf RL agent in an appendix for the camera-ready version of the paper.

---

> ### Author Response · Authors · 2023-11-22
> **Re: computational resources**
>
> > The computational resource requirements for training RECODE remain unclear. The paper indicates a substantial number of environment interactions, around 7.5e8 steps, for various environments, comparable to previous studies. However, essential details such as memory usage, CPU requirements, and GPU memory are not provided. This lack of information poses a significant concern, as it could either limit the framework's practical applicability or potentially underscore its uniqueness as a noteworthy contribution.
>
> We updated the submission. For each of the experiments we now report in App. F total resources used to generate one run (i.e. one seed) across the whole distributed agent (including multiple actors, learner and RECODE memory mechanism, see App. H for more details).
>
> One seed for an Atari experiment (MEME-RECODE-AP MEME-NGU-AP) took 24h to execute using multiple servers with a total of 64 CPUs, 1TB RAM, and 5 TPUv4.
> One seed for a DM-HARD-8 experiment (both MEME-RECODE-CASM and MEME-NGU-CASM) took 90h to execute using multiple servers with a total of 512 CPUs, 1TB RAM, and 5 TPUv4.

---

### Official Review · Reviewer_BYvq · 2023-10-31

**Soundness:** 2 fair
**Presentation:** 2 fair
**Contribution:** 2 fair
**Rating:** 6
**Confidence:** 4

**Summary:**

The document presents a novel approach known as RECODE for enhancing exploration in the realm of Reinforcement Learning. This method combines elements from both parametric and non-parametric techniques in this domain.

Initially, the paper delves into the existing literature of novelty-driven exploration. Parametric strategies involve assessing novelty by utilizing trained representation models on states and actions. However, these methods are prone to issues like catastrophic forgetting or predicting novelty where it is irrelevant with respect to "controllable" features. On the other hand, non-parametric techniques maintain a repository of historical embeddings. They employ approaches such as state counting to drive novelty-based exploration, or approximate state counting in continuous domains.

In contrast to traditional methods, the authors introduce RECODE, which is a non-parametric approach relying on a historical embedding repository. Instead of simply erasing entries from this history, RECODE dynamically adjusts embeddings and counters by employing a system of discounted sums.

The paper also offers a series of experiments showcasing RECODE's superior performance when compared to baseline methods. Remarkably, RECODE stands out as the first Reinforcement Learning algorithm to tackle the game "Pitfall!" and surpass the human baseline in "Push Blocks" from DM-HARD-8.

**Strengths:**

The paper addresses an important problem within the realm of Reinforcement Learning by introducing an algorithm centered on novelty-driven exploration. As a solution, RECODE effectively mitigates significant shortcomings observed in prior visitation count techniques like NGU. It strikes a balance between short-term and long-term novelty, avoiding an excessive bias toward short-term novelty and integrating long-term memory into the novelty determination process, despite having finite memory constraints.

In the experiments, the RECODE approach completes the “Pitfall!” benchmark for the first time, and it also attains performance levels exceeding human capabilities on the "Push Blocks" challenge from DM-HARD-8.

**Weaknesses:**

The rationale behind using RECODE as opposed to NGU is simplicity: RECODE uses one singular mechanism, while NGU uses two. However, it is not apparent to me that overall RECODE is simpler in terms of hyperparameter count / design space. Is there a rationale behind the optimal choice of hyperparmaters?

In addition, the experiment figures do not seem to support an overwhelming positive conclusion in favor of RECODE, besides for the headline successes on “Pitfall!” and DM-HARD-8.

**Questions:**

Would you kindly answer the following questions?
* Please clarify how the experimental figures support the conclusion that the method is indeed superior to alternatives.
* I believe that Figure 1 would benefit from a more explanatory caption. There is no definition for HNS, Capped HNS and the number of frames in the caption.
* Could you explain how your method is better than the alternatives when it comes to design space complexity?

---

> ### Author Response · Authors · 2023-11-22
> **Re: Could you explain how your method is better than the alternatives when it comes to design space complexity?**
>
> Note that when compared to NGU, RECODE:
> - Removes the complexity of having to rely on a parametric novelty detector (RND) for long-term memory, which involved hard to tune hyperparameters and design choices including selecting a network architecture and network optimizer, and how to combine the RND reward and episodic reward
> - Inherits the following hyperparameters from the episodic component of NGU: atoms removal, reward constant, decay rate, # nearest neighbors, and relative tolerance (referred to as cluster distance in that work)
> - Adds hyperparameters for discount, insertion probability, memory update and memory size
>
>
> Taking this into consideration the slight increase in complexity of the memory mechanism is more than offset by the removal of RND, and overall we consider RECODE a less complex novelty detection mechanism (both conceptually and in ease of tuning) than NGU.
>
> Ensuring that the reader can clearly understand which of these new hyperparameters are important to tune and when remains important. We have added some general guidelines in Section 3 of the paper, as well as an ablation over memory size and insertion probability in Appendix L.1, Fig. 20.
>
> Notice also that in general most of RECODE's design space (memory upkeep, representation learning, forgetting) is shared across the family of non-parametric visitation counts estimators (that goes beyond EMM/NGU). More in detail, comparing RECODE to its closest alternative NGU we have that:
> - Memory size: NGU side-steps tuning a memory size by implicitly setting the EMM size to be the maximum length of an episode. Note that this is not possible in many cases (long episodes cannot be all stored in RAM), and makes it impossible to preserve long-term memory. To cover these limitations, NGU introduces extra complexity with a completely separate long-term novelty detector (RND). Therefore, both RECODE and NGU are more complex than vanilla EMM: RECODE through the memory management methods with their hyperparameters; and NGU through RND with its hyperparameters that need to be tuned (network architecture, optimizer, etc…). Overall we find that RECODE is robust when choosing its new hyperparameters (see updated ablations after rebuttal). In contrast we found that RND is quite difficult to tune and has a complex interaction with the episodic reward and representation learning which makes it more difficult to reason about. Removing RND makes it easier to tune the overall novelty mechanism.
> - Heuristics to update/add/remove atoms of memory: this is an inherent feature of all finite-size memories, since choosing a removal strategy is always necessary when dealing with finite memory and potentially infinite interactions. NGU’s choice in this regard seems simple (add every atom encountered and remove them all at the end of the episode), but it is not applicable to complex real-world scenarios (e.g. if the episode length is larger than what is computationally feasible to store in memory we *must* remove atoms before the end of an episode). Overall, NGU can avoid memory management heuristics because all the heuristics are pushed into the RND module, while RECODE must explicitly address them. Compared to existing novelty estimation methods for RL our discounting and cluster merging approaches are more novel, but they are still rooted in and inspired by a long tradition of non-stationary clustering methods such as DP-Means (see App. D). Similarly, our particular design choice for removal was motivated by empirical observations in toy experiments (see App. D), but as stated, it is ultimately an arbitrary choice. Our ablation against alternative removal strategies demonstrated that RECODE is robust to this choice.
> - Discounting: $\gamma$ is one of the two important hyperparameters that we recommend should be tuned in the guidelines we have added in the latest version. We give more insight on how it impacts RECODE’s performance in Appendix D  including illustrative toy experiments, theoretical connections to DP-Means and ablations on the same complex environments used in the main paper. In particular, we show that the usefulness and impact of $\gamma$ strongly depends on the non-stationarity of the observations, which in turns depend on both the evolution of the observation representation and the diversity of the data collected by the agent.
> - Representation learning method (i.e. embedding) and the associated distance kernel were also important design choices in prior non-parametric methods. The adaptive kernel we use is also the same as that used in NGU, except for the modification to sum over all atoms within an epsilon ball of the query embedding rather than over a fixed number of nearest neighbours. This complication was unnecessary in NGU because every atom represented a count of 1 and thus did not suffer from the undesirable behaviour that adding an embedding could decrease the count for that embedding when using the latter approach (see discussion under Eq. 3)

---

> ### Author Response · Authors · 2023-11-22
> **Re: Figure 1's caption**
>
> Human normalized score (HNS) is introduced at the end of Sec. 1 where Fig. 1 is first referenced, as well as in the appendix. We have also include appropriate references and definitions in the caption for completeness in the updated version of the paper.

---

> ### Author Response · Authors · 2023-11-22
> **Re: Please clarify how the experimental figures support the conclusion that the method is indeed superior to alternatives.**
>
> While all reviewers agree that the our main claims (SOTA performance on the harder DM-HARD-8 task and novel capabilities in "Pitfall!"), they also have some questions on the Atari experiments.
> Our main intention on Atari is to demonstrate that RECODE is in-line with the current state-state-of-the-art achieved by MEME-NGU, since performance on most games already far exceeds the human benchmark and has limited room for improvement. To that end we point out that mean and median Human Normalized Score (HNS) are higher for RECODE, as is the mean capped HNS. In addition RECODE achieves better or equal mean final performance and AUC in 6/8 games (vs 3/8 and 2/8 respectively).
>
> That being said, ​​the difference under these metrics for most games is small, and only statistically significant (to 5% significance level) in a few cases. Specifically, when using a one-sided Mann-Whitney U test for difference in mean across seeds we observe the following: RECODE > NGU in Pitfall (p=0.0043) and Q*Bert (p=0.0465) for final performance and in Solaris (p=0.0011) and Seaquest (p=0.0325) for AUC. Conversely, the only case where NGU > RECODE is final performance for Hero (p=0.0465). For aggregate statistics we can compute p-values via a bootstrap estimate and find that RECODE > NGU for Mean HNS (p=0.0185) and Mean Capped HNS (p=0.0419). We will modify the wording in the main text to avoid possible misinterpretations and include an appendix summarizing all of these metrics with appropriate confidence intervals and hypothesis tests (see Table 6 in the revised appendix).

---

### Official Review · Reviewer_tYtD · 2023-10-31

**Soundness:** 3 good
**Presentation:** 3 good
**Contribution:** 3 good
**Rating:** 8
**Confidence:** 4

**Summary:**

This paper proposes a novelty-based intrinsic reward (RECODE) to facilitate exploration in sparse-reward environments. The proposed approach derives intrinsic rewards using approximate state visitation counts in a suitably constructed embedding space.

RECODE builds on the previously proposed Never Give Up (NGU) intrinsic reward in two ways.

First, unlike in NGU, which uses one-step inverse dynamics to learn an observation embedding, this paper proposes Coupled Action-State Masking (CASM) to capture controllable features across multiple steps.

Second, the proposed approach maintains and derives intrinsic rewards from a global memory of embeddings updated based on a clustering principle. Whereas NGU derived intrinsic rewards using the contents of an episodic buffer and a global bonus from Random Network Distillation (RND).

Experiments show that the proposed approach to providing intrinsic rewards outperforms baselines in challenging exploration settings. Further analysis shows that the proposed approach is more robust to observation noise than NGU and RND.

**Strengths:**

**S1.**  The problem of exploration in sparse-reward settings and large observation spaces is of significant interest to the research community.

**S2.** The proposed approach simplifies NGU in a crucial way. NGU needed to combine an episodic bonus and a global bonus from RND. RECODE removes the need for a separate episodic novelty component. However, this simplification comes at the cost of mechanisms to manage the memory (see W1).

**S3.** In terms of the empirical evaluation, the environments considered are indeed challenging exploration problems. Further, the proposed approach provides empirical benefits, especially on the DM-HARD-8 problems.

**S4.** The paper is well-written and easy to follow.

**Weaknesses:**

**W1.** A crucial weakness is that the proposed approach introduces additional complexity in memory management compared to NGU, which had a more straightforward episodic reset for memory. Here, there are hyperparameters for atoms of memory size, a discounting of counts and additional heuristics to update/add/remove atoms of memory. Some of the additional complexity and increase in hyperparameters might make it hard to apply this approach to new environments.

**W2.** Further ablations are needed to clarify certain issues.

Is CASM only beneficial over one-step action prediction (AP) when there is high aliasing due to partial observability (like DM-HARD-8)? Are results for MEME-RECODE/NGU-CASM available for the Atari environments? I may have missed them in the appendix.

In Appendix L, results are presented for RND on AP (one-step action prediction), where a random net is applied to features extracted by AP. As the random network is typically seen as a feature extractor itself, wouldn’t it be more natural to obtain RND-like intrinsic rewards from a predictor of the AP feature of a state (rather than a random embedding of the AP embedding)?

Similarly, it would be helpful to evaluate if RECODE is better than NGU when NGU uses RND on AP (or an RND-like bonus with AP features directly) for its global bonus.

**W3.** The paper would benefit from connections to existing work. For instance, the recently proposed MIMEx [1] also uses masked transformers to derive trajectory level intrinsic rewards.

Previous works have studied incorporating information over longer trajectories to derive better intrinsic rewards under partial observability, [4] uses general value functions based on RND, other approaches use successor features which naturally incorporate multi-step information [2,3].

**W4.** In the abstract and the conclusion, the authors claim that the proposed approach sets a new state-of-the-art on Atari’s hard exploration dataset and DM-HARD-8. While this claim is reasonable for DM-HARD-8, I am not sure it applies to the Atari experiments, as RECODE appears to be worse than NGU in hero and venture.

I remain open to increasing my score, should the weaknesses and questions be adequately addressed/clarified.


—------------------—------------------—------------------—------------------—------------------

### References

[1] Lin, T., & Jabri, A. (2023). MIMEx: Intrinsic Rewards from Masked Input Modeling. arXiv preprint arXiv:2305.08932

[2] Machado, M. C., Bellemare, M. G., & Bowling, M. (2020). Count-based exploration with the successor representation. In Proceedings of the AAAI Conference on Artificial Intelligence

[3] Janz, D., Hron, J., Mazur, P., Hofmann, K., Hernández-Lobato, J. M., & Tschiatschek, S. (2019). Successor uncertainties: exploration and uncertainty in temporal difference learning. Advances in Neural Information Processing Systems

[4] Ramesh, A., Kirsch, L., van Steenkiste, S., & Schmidhuber, J. (2022). Exploring through random curiosity with general value functions. Advances in Neural Information Processing Systems

------------------------------------------------------------------------------------------------------

**EDIT (Post-rebuttal):**

Thanks for your detailed response to the weaknesses and for sharing your thoughts regarding resets in different procedurally generated environments. The response largely addresses my concerns, and I have updated my score accordingly.

The promised experiments/ablations and clarification about RND-on-AP alleviate my concerns about W2. Incorporating the comments shared in the reply to W3 would help better contextualize the contributions of RECODE. Modifying the wording and including the details shared here will avoid misinterpreting claims (W4).

Coming back to W1, your response helps me better understand the "complexity" trade-off with respect to NGU. I appreciate the point regarding long episodes. Removing RND is a significant plus, especially if it is hard to tune. The new sensitivity analysis in the appendix supports the paper (it would be nice to see more values considered in the next version). *Ideally*, this claim should also be backed through sensitivity analysis with NGU/RND.

A minor note I missed in the original review is a slight inconsistency between the Background and General Notation sections. For example, policies map observations to distribution over actions in Background, but in General Notation, the policy operates on histories.

Another minor issue is that the caption of Figure 6 uses top/bottom instead of left/right.

**Questions:**

Q. In many procedurally generated environments, episodic resets to the memory (as in NGU) could be preferable. Consider a scenario where blue circles are actually novel in the current episode (and should be sought) but have been seen in previous episodes in other contexts. Of course, some notion of global novelty would also typically be needed. It would seem that something like NGU would again be preferable to RECODE in many of these settings. I am curious to know the authors’ thoughts regarding this.

---

> ### Author Response · Authors · 2023-11-22
> **Adressing open questions**
>
> We would like to thank the reviewer for the very detailed feedback. We hope our replies adequately address your concerns and give you the confidence to raise your score
>
> > Q. In many procedurally generated environments [...]
>
> This is an important question that we have indeed given some thought to. The answer to whether or not hard resets could be preferable depends on both the nature and distribution of procedurally generated elements, as well as the choice of representation used in conjunction with RECODE.
>
> In particular if we consider the example the reviewer mentions (known object in a new context) a “good” learned representation will be capable of capturing this novel context/combination as a novel embedding that will trigger high reward in RECODE. Empirically we note that RECODE-CASM seems to achieve this desirable behaviour, and performs well in DM-HARD-8 which already contains the following procedurally generated elements: initial position and orientation of the agent; shapes, colors, textures, and positions of objects; wall and floor colors and textures, and sensor colors (which can only be activated by an object of the matching color). Following again from the example, the agent learns a representation where a blue ball will mean different things in different contexts and correctly assign novelty. It is also plausible that the action-prediction objective aliases together some of the procedural variability between episodes, though we have not tested this specifically.
>
> On the other hand we would expect that more complex procedurally generated elements such as level geometry and size may be more challenging for generalization across episode boundaries, particularly when the degree of variability is very large. While it would be possible to combine just the episodic memory component of NGU, with RECODE used in place of RND for long-term memory, this solution is not ideal due to the added complexity of implementation and tuning. An simpler approach could be to investigate how to construct representations that are suitable for generalization across episode boundaries in this setting. For example, it is plausible that higher-level semantic representations, or behavior representations can still generalize well in this setting, and we believe these could be promising directions for future research.

---

> ### Author Response · Authors · 2023-11-22
> **Re: W1 additional complexity in memory management compared to NGU**
>
> Note that when compared to NGU, RECODE:
> - Removes the complexity of having to rely on a parametric novelty detector (RND) for long-term memory, which involved hard to tune hyperparameters and design choices including selecting a network architecture and network optimizer, and how to combine the RND reward and episodic reward
> - Inherits the following hyperparameters from the episodic component of NGU: atoms removal, reward constant, decay rate, # nearest neighbors, and relative tolerance (referred to as cluster distance in that work)
> - Adds hyperparameters for discount, insertion probability, memory update and memory size
>
>
> Taking this into consideration the slight increase in complexity of the memory mechanism is more than offset by the removal of RND, and overall we consider RECODE a less complex novelty detection mechanism (both conceptually and in ease of tuning) than NGU.
>
> Ensuring that the reader can clearly understand which of these new hyperparameters are important to tune and when remains important. We have added some general guidelines in Section 3 of the paper, as well as an ablation over memory size and insertion probability in Appendix L.1, Fig. 20.
>
> Notice also that in general most of RECODE's design space (memory upkeep, representation learning, forgetting) is shared across the family of non-parametric visitation counts estimators (that goes beyond EMM/NGU). More in detail, comparing RECODE to its closest alternative NGU we have that:
> - Memory size: NGU side-steps tuning a memory size by implicitly setting the EMM size to be the maximum length of an episode. Note that this is not possible in many cases (long episodes cannot be all stored in RAM), and makes it impossible to preserve long-term memory. To cover these limitations, NGU introduces extra complexity with a completely separate long-term novelty detector (RND). Therefore, both RECODE and NGU are more complex than vanilla EMM: RECODE through the memory management methods with their hyperparameters; and NGU through RND with its hyperparameters that need to be tuned (network architecture, optimizer, etc…). Overall we find that RECODE is robust when choosing its new hyperparameters (see updated ablations after rebuttal). In contrast we found that RND is quite difficult to tune and has a complex interaction with the episodic reward and representation learning which makes it more difficult to reason about. Removing RND makes it easier to tune the overall novelty mechanism.
> - Heuristics to update/add/remove atoms of memory: this is an inherent feature of all finite-size memories, since choosing a removal strategy is always necessary when dealing with finite memory and potentially infinite interactions. NGU’s choice in this regard seems simple (add every atom encountered and remove them all at the end of the episode), but it is not applicable to complex real-world scenarios (e.g. if the episode length is larger than what is computationally feasible to store in memory we *must* remove atoms before the end of an episode). Overall, NGU can avoid memory management heuristics because all the heuristics are pushed into the RND module, while RECODE must explicitly address them. Compared to existing novelty estimation methods for RL our discounting and cluster merging approaches are more novel, but they are still rooted in and inspired by a long tradition of non-stationary clustering methods such as DP-Means (see App. D). Similarly, our particular design choice for removal was motivated by empirical observations in toy experiments (see App. D), but as stated, it is ultimately an arbitrary choice. Our ablation against alternative removal strategies demonstrated that RECODE is robust to this choice.
> - Discounting: $\gamma$ is one of the two important hyperparameters that we recommend should be tuned in the guidelines we have added in the latest version. We give more insight on how it impacts RECODE’s performance in Appendix D  including illustrative toy experiments, theoretical connections to DP-Means and ablations on the same complex environments used in the main paper. In particular, we show that the usefulness and impact of $\gamma$ strongly depends on the non-stationarity of the observations, which in turns depend on both the evolution of the observation representation and the diversity of the data collected by the agent.
> - Representation learning method (i.e. embedding) and the associated distance kernel were also important design choices in prior non-parametric methods. The adaptive kernel we use is also the same as that used in NGU, except for the modification to sum over all atoms within an epsilon ball of the query embedding rather than over a fixed number of nearest neighbours. This complication was unnecessary in NGU because every atom represented a count of 1 and thus did not suffer from the undesirable behaviour that adding an embedding could decrease the count for that embedding when using the latter approach (see discussion under Eq. 3)

---

> ### Author Response · Authors · 2023-11-22
> **Re: W2 Further ablations**
>
> > Is CASM only beneficial over one-step action prediction (AP) when there is high aliasing due to partial observability (like DM-HARD-8)? Are results for MEME-RECODE/NGU-CASM available for the Atari environments? I may have missed them in the appendix.
>
> Results for MEME-RECODE/NGU-CASM on Atari were not included in the initial submission but we have started running these experiments and will include them in the camera-ready version of the paper. While we do not yet have results for all seeds and games, preliminary results suggest that performance is very similar to the AP embeddings on Atari. Indeed, we would suspect that the primary benefit of CASM comes from application to domains with a high degree of partial observability, and our early results would seem to support that hypothesis.
>
> > In Appendix L, results are presented for RND on AP (one-step action prediction), where a random net is applied to features extracted by AP. As the random network is typically seen as a feature extractor itself, wouldn’t it be more natural to obtain RND-like intrinsic rewards from a predictor of the AP feature of a state (rather than a random embedding of the AP embedding)?
> Similarly, it would be helpful to evaluate if RECODE is better than NGU when NGU uses RND on AP (or an RND-like bonus with AP features directly) for its global bonus.
>
> Assuming the AP embedding network is already trained, it could make sense to apply a prediction based reward leveraging it directly as the reviewer suggests. However for the setting we focus on (where the AP embedding network is concurrently trained with the RL network) it seems likely that such a predictor network would just track the AP embedding network closely and thus may not confer any benefit over just using the AP loss itself to construct an intrinsic reward. A high AP loss may be the result of states being novel, but will also happen if many actions lead to the same state transition; which might have the undesirable effect of incentivizing the agent to seek out uncontrollable states. A good example of this occurs in Pitfall; where after jumping, no action will alter the character’s trajectory until the jump has landed.
>
> Additionally, one of the intuitions behind RND is that the random network should not generalize across states, otherwise states which are actually novel but have similar features to frequently observed states may have low reward. Applying a random nonlinear transformation on top of the AP embedding features is intended to reduce this type of inappropriate generalization, and this very feature explains why AP-on-RND performs poorly when the AP embedding network is concurrently trained.
>
> Since we observe that RND-on-AP only helps when using a pretrained AP embedding network, it may make sense to directly compare RECODE and NGU with RND-on-AP in this specific setting, and we can commit to include this experiment in the camera-ready version of the paper.

---

> ### Author Response · Authors · 2023-11-22
> **Re: W3 connections to existing work**
>
> We thank the referee for the proposed references, which we will add in the related works section together with the other connections to previous works on novelty-based intrinsic motivation already present. Looking more closely at each of the suggestions:
>
> [1] MIMEx uses the loss of a masked transformer as an intrinsic reward signal. This masked transformer acts on a latent space, which is built on top of an initial fully-connected embedding layer applied on the input. The masked transformer thus estimates novelty according to the prediction error in a latent space.
> Our approach disentangles the creation of a meaningful embedding and the intrinsic reward signal. In particular, we use a masked transformer (CASM in our paper) to build a robust latent space. Indeed, it is well-known in literature that action-prediction based embeddings are usually more robust to noise, as they tend to ignore uncontrollable factors in the input. Being the output of a neural network, in MIMEx the reward signal will change slowly and might not capture episodic novelty, while also being vulnerable to catastrophic forgetting. On the other hand, we don’t use the masked transformer loss as an intrinsic reward signal, but we use the RECODE component. As described in the introduction of the paper, this allows us to extend the advantages of short-term intrinsic reward techniques (which previously were essentially episodic) to longer horizons, while avoiding slow adaptation and catastrophic forgetting issues that are often connected to the use of a neural network for the reward signal.
>
> We further note that MIMEx uses a sequence auto-encoding objective operating only on states, whereas CASM instead predicts actions, while operating on both states and actions; with the masking strategy being designed so that exactly one of the action and state are observed at any given timestep, so as to better preserve the requisite information needed for the prediction task.
>
> [4] develops the idea of using general value functions to take into account the prediction error on more than a single step. In this way, they couple the prediction task to the current policy, giving it extra incentive for explorative behaviour. This technique is complementary to our novelty-estimation technique (RECODE), as one could also build a value function with it to explore over longer horizons. In addition, their state representation is based on a random network, which might suffer from the presence of noisy observations, and thus is expected to have limited applicability when environments are highly stochastic or contain uncontrollable features.
>
> [2,3] Successor states have been used as an approximation of state visitation count, while successor uncertainties can provide an intrinsic reward signal based on prediction uncertainty. However, those works estimate visitation counts directly in the state space. We emphasize the importance of building an insightful representation in the case of complex environments, where the state space can be extremely large and possibly contain many irrelevant features, thus requiring some type of aggregation or filtering mechanism. In those cases, a good representation is needed to filter out noise and environmental stochasticity. In particular, an efficient exploration strategy should be biased towards controllable novel states, and this is why we focus on action-prediction representations. The proposed multi-step action-prediction (CASM) proved to provide a more robust representation with respect to the competing representation learning techniques. Successor features (Successor Features for Transfer in Reinforcement Learning, arXiv:1606.05312) introduces in the embedding space the same approach of successor states as approximation of novelty.
>
> We note that [1] & [4] evaluate on a different set of tasks so direct comparison is difficult. [2] & [3], and the works extending successor states to successor features, do have some overlap with the Atari games we evaluate, but underperform compared to methods such as RND, NGU, or pseudo-count based rewards on both Montezuma's Revenge or Pitfall which are generally regarded as the most challenging environments for exploration in the Atari57 task suite.

---

> ### Author Response · Authors · 2023-11-22
> **Re: W4 new state-of-the-art on Atari**
>
> As we describe in the general response, our main intention on Atari is to demonstrate that RECODE is in-line with the current state-state-of-the-art achieved by MEME-NGU, since performance on most games already far exceeds the human benchmark and has limited room for improvement. To that end we point out that mean and median Human Normalized Score (HNS) are higher for RECODE, as is the mean capped HNS. In addition RECODE achieves better or equal mean final performance and AUC in 6/8 games (vs 3/8 and 2/8 respectively).
>
> That being said, ​​the difference under these metrics for most games is small, and only statistically significant (to 5% significance level) in a few cases. Specifically, when using a one-sided Mann-Whitney U test for difference in mean across seeds we observe the following: RECODE > NGU in Pitfall (p=0.0043) and Q*Bert (p=0.0465) for final performance and in Solaris (p=0.0011) and Seaquest (p=0.0325) for AUC. Conversely, the only case where NGU > RECODE is final performance for Hero (p=0.0465). For aggregate statistics we can compute p-values via a bootstrap estimate and find that RECODE > NGU for Mean HNS (p=0.0185) and Mean Capped HNS (p=0.0419). We will modify the wording in the main text to avoid possible misinterpretations and include an appendix summarizing all of these metrics with appropriate confidence intervals and hypothesis tests (see Table 6 in the revised appendix).

---

### Author Response · Authors · 2023-11-22
**Addressing shared concerns**

We thank the reviewers for the valuable feedback, which we have began to incorporate in our revised submission. There were two concerns shared among multiple reviewers that we address here in general, and more in detail in each reviewer’s answer.



**Selecting appropriate hyperparameters for RECODE**

We will improve the discussion of the hyperparameters and already include in the updated main paper (end of section 3) simple guidelines indicating that in practice only two hypers should be slightly tuned across experiments (discount $\gamma$, and memory size), while the others are robust across a wide range of values and environments. To support these guidelines we refer the reviewers (and in future the readers) to the appropriate appendix sections for an in-depth ablation and analysis.

In particular,  App. F discusses the hypers that were tuned across experiments, where we can see that only memory size, $\gamma$ and $\eta$ are changed across experiments, while $\kappa, c, \tau$ and $k$ are kept constant across experiments and can be considered a reliable default. In the final version we will further strengthen this statement by providing results for all the values we swept over during the hyperparameter search. We already updated the paper (see new expanded App. L) to include the first of these additional ablations showing that RECODE is also robust to changes in $\eta$, and memory size, always exceeding human baseline. This will complement the App. D. ablations on removal strategy and discount factor, and fully support the guidelines.



**Regarding the claim that RECODE achieves SOTA performance on Atari**

While all reviewers agree that our main claims (SOTA performance on the harder DM-HARD-8 task and novel capabilities in "Pitfall!") are well supported, they also have some questions on the Atari experiments.
Our main intention on Atari is to demonstrate that RECODE is in-line with the current state-state-of-the-art achieved by MEME-NGU, since performance on most games already far exceeds the human benchmark and has limited room for improvement. To that end we point out that mean and median Human Normalized Score (HNS) are higher for RECODE, as is the mean capped HNS. In addition RECODE achieves better or equal mean final performance and AUC in 6/8 games (vs 3/8 and 2/8 respectively).

That being said, ​​the difference under these metrics for most games is small, and only statistically significant (to 5% significance level) in a few cases. Specifically, when using a one-sided Mann-Whitney U test for difference in mean across seeds we observe the following: RECODE > NGU in Pitfall (p=0.0043) and Q*Bert (p=0.0465) for final performance and in Solaris (p=0.0011) and Seaquest (p=0.0325) for AUC. Conversely, the only case where NGU > RECODE is final performance for Hero (p=0.0465). For aggregate statistics we can compute p-values via a bootstrap estimate and find that RECODE > NGU for Mean HNS (p=0.0185) and Mean Capped HNS (p=0.0419). We will modify the wording in the main text to avoid possible misinterpretations and include an appendix summarizing all of these metrics with appropriate confidence intervals and hypothesis tests (see Table 6 in the revised appendix).

---

### Meta-Review · Area_Chair_G8xW · 2023-12-24

**Metareview:**

### Summary
The paper presents proposes a novelty-based intrinsic reward to facilitate exploration in sparse-reward environments.
The proposed approach derives intrinsic rewards using approximate state visitation counts in an embedding space. The framework consists of two modules:
	- RECODE, a memory buffer that approximates visitation counts in a learned embedding space; and
	- CASM, a transformer model that learns the embedding space by jointly addressing masked sequence modelling and inverse dynamics prediction.


###  Strengths
+ exploration in sparse-reward settings and large observation spaces
+ Strong Experimental evaluation as well as comparisons to baselines (NGU and BYOL-Explore)
+ Surprising conclusion: "Learning predictions is sufficient to induce useful structure into representations used by other regions."

### Weaknesses:
- Experimental complexity is limited to environments only requiring intrinsic motivation. Experimental evaluation on the interplay and marginal benefits of exploration in domains like Minecraft will be very informative.
- Additional complexity and hyperparams in memory mechanism. This requires appropriate choice of hyperparams, a concern expressed by multiple reviewers.

**Justification For Why Not Higher Score:**

Reviewers provide a high score, but limited experimental complexity limits the scope of claims.

**Justification For Why Not Lower Score:**

All Reviewers are in agreement of the novelty and technical value of the method.

---

### Decision · Program_Chairs · 2024-01-16

Accept (spotlight)